# The mechanism of kinesin inhibition by kinesin-binding protein

**Joseph Atherton[1,2]\*, Jessica JA Hummel[3], Natacha Olieric[4], Julia Locke[2†], Alejandro Peña[2‡], Steven S Rosenfeld[5], Michel O Steinmetz[4,6], Casper C Hoogenraad[3], Carolyn A Moores[2]**

[1]Randall Centre for Cell and Molecular Biophysics, King's College, London, United Kingdom; [2]Institute of Structural and Molecular Biology, Birkbeck, University of London, London, United Kingdom; [3]Cell Biology, Neurobiology and Biophysics, Department of Biology, Faculty of Science, Utrecht University, Utrecht, Netherlands; [4]Laboratory of Biomolecular Research, Division of Biology and Chemistry, Paul Scherrer Institut, Villigen PSI, Switzerland; [5]Department of Cancer Biology, Mayo Clinic, Jacksonville, United States; [6]University of Basel, Biozentrum, Basel, Switzerland

**Abstract** Subcellular compartmentalisation is necessary for eukaryotic cell function. Spatial and temporal regulation of kinesin activity is essential for building these local environments via control of intracellular cargo distribution. Kinesin-binding protein (KBP) interacts with a subset of kinesins via their motor domains, inhibits their microtubule (MT) attachment, and blocks their cellular function. However, its mechanisms of inhibition and selectivity have been unclear. Here we use cryo-electron microscopy to reveal the structure of KBP and of a KBP–kinesin motor domain complex. KBP is a tetratricopeptide repeat-containing, right-handed α-solenoid that sequesters the kinesin motor domain's tubulin-binding surface, structurally distorting the motor domain and sterically blocking its MT attachment. KBP uses its α-solenoid concave face and edge loops to bind the kinesin motor domain, and selected structure-guided mutations disrupt KBP inhibition of kinesin transport in cells. The KBP-interacting motor domain surface contains motifs exclusively conserved in KBP-interacting kinesins, suggesting a basis for kinesin selectivity.

**\*For correspondence:**
joseph.atherton@kcl.ac.uk

**Present address:** [†]Macromolecular Machines Laboratory, The Francis Crick Institute, London, United Kingdom; [‡]Department of In Silico Drug Discovery, Pharmidex Pharmaceuticals, Hatfield, United Kingdom

**Competing interests:** The authors declare that no competing interests exist.

## Introduction

Kinesins are a superfamily of microtubule (MT)-based molecular motors that play important roles in cellular functions such as mitosis, cell motility, and intracellular transport (*Vale, 2003*; *Hirokawa et al., 2009*; *Klinman and Holzbaur, 2018*). Kinesins are categorised into 14 sub-classes (kinesin-1 to kinesin-14 [*Lawrence et al., 2004*]) by motor domain conservation and within these sub-classes individual family members (a total of 45 'KIF' or 'Kif' genes in humans and mice respectively) have a wide range of functional characteristics and biological roles (*Vale, 2003*; *Miki et al., 2001*). Dysfunction of kinesin family members has been implicated in a number of pathological conditions (*Hirokawa et al., 2010*; *Mandelkow and Mandelkow, 2002*). The kinesin motor domain is the MT-binding engine that drives these activities, converting the chemical energy of ATP binding and hydrolysis into mechanical force. While these mechanical forces are classically used to generate motility in transport kinesins, some kinesin family members drive MT organisation or depolymerisation of MTs.

Kinesins are highly regulated in order to prevent both waste of ATP and to spatially and temporally control kinesin function. This is particularly important in highly polarised and compartmentalised cells such as neurons. Kinesin regulation via inhibition of their motor domains can occur through a number of mechanisms that limit ATPase activity and/or block track binding – these include

intramolecular inhibition by kinesin tail domains, post-translational modification of the motor, or through interactions with regulatory binding partners. Recently, it has been demonstrated that a subset of kinesin superfamily members, including kinesin-2s, −3 s, −8 s, and −12 s, are sequestered by kinesin-binding protein (KBP; KIF1BP; KIAA1279), which inhibits MT track attachment by their motor domains and, thus, blocks their MT-related functions (*Wozniak et al., 2005*; *Kevenaar et al., 2016*; *Alves et al., 2010*).

KBP is expressed in multiple human tissues including brain and heart (*Wozniak et al., 2005*). Mutations in the KBP have been identified as causing autosomal recessive Goldberg-Shprintzen syndrome (GOSHS) (*Brooks et al., 2005*; *Dafsari et al., 2015*; *Valence et al., 2013*; *Salehpour et al., 2017*), which presents as congenital facial dysmorphia, nervous system pathology, and dysfunction and heart defects (*Tanaka et al., 1993*). In addition, KBP gene copy number has been recently reported as predictive in paediatric neuroblastoma prognosis, prompting its suggestion as a drug target (*Suo et al., 2018*). KBP was originally identified as a kinesin-3-binding partner that modulated its mitochondrial transport function (*Wozniak et al., 2005*); however, KBP has since been shown to interact with a subset of other kinesin family members to regulate diverse cellular processes including mitosis (*Brouwers et al., 2017*; *Malaby et al., 2019*), spermatogenesis (*Lehti et al., 2015*), and neuronal differentiation, growth, and cargo distribution (*Alves et al., 2010*; *Lyons et al., 2008*; *Drévillon et al., 2013*; *Drerup et al., 2016*; *Chang et al., 2019*; *Hirst et al., 2017*).

We do not currently know what the structure of KBP is, nor understand the mechanism of KBP–kinesin inhibition. It is also completely unknown how KBP differentiates between particular kinesin family members. KBP is a 72 kDa protein, is predicted to contain several tetratricopeptide repeats (TPRs), and to be mainly α-helical in secondary structure content (*Wozniak et al., 2005*; *Kevenaar et al., 2016*). Here we present cryo-electron microscopy (cryo-EM) structures of KBP alone and of KBP bound to the motor domain of the human mitotic kinesin KIF15 (a 110 kDa complex). We show that KBP is a TPR-containing, right-handed α-solenoid protein composed of nine antiparallel α-helix pairs interrupted by a linker region. We also show that KBP's concave face binds KIF15 via its MT-binding elements and induces a large displacement of the kinesin α4 helix, sterically inhibiting MT association. Finally, we show that KBPs kinesin selectivity is associated with specific kinesin sequences spread across the interaction surface.

## Results

### KBP is a TPR-containing, right-handed α-solenoid

The 3D structure of the ~72 kDa KBP at 4.6 Å resolution (*Figure 1* and *Figure 1—figure supplement 1a and b*) was determined using cryo-EM data collected using a Volta phase plate (VPP), and an atomic model was calculated (see Materials and methods *Table 1*). Our structure revealed that KBP is a right-handed α-solenoid protein (*Figure 1a and b* and *Figure 1—figure supplement 1c–e*). Nine pairs of anti-parallel α-helices (αHP1 [α-helical pair 1] to αHP9) are broken by a single 'linker α-helix' (LαH) and 'linker loop' (LL) in the centre of the fold separating KBP into N-terminal and C-terminal subdomains (*Figure 1*, *Figure 1—figure supplement 1c–e*, and *Figure 1—figure supplement 2*). The four predicted TPR motifs contribute exclusively to α-helical pairs in the N-terminal subdomain (*Figure 1a,d, and e*).

The supercoiling α-helical pairs form concave and convex faces linked by short and long loops that constitute the two edges of the α-solenoid (*Figure 1—figure supplement 1c–e* and *Figure 1—figure supplement 2*). In contrast to the shorter loops, the longer loops (more than seven residues) tend to be partially disordered, show low sequence homology between KBP orthologues in different species, and are mainly found in the N-terminal subdomain (e.g. L2, L6, and L10; *Figure 1—figure supplement 1d and e* and *Figure 1—figure supplement 2*). The LL is the longest (62 residues) and is thus unique in the KBP structure because it is reasonably conserved and mainly ordered, with visible corresponding density clearly bridging the N- and C-terminal subdomains (*Figure 1a and b*, *Figure 1—figure supplement 1d and e*, *Figure 1—figure supplement 2*, and *Figure 1—figure supplement 3*). Despite this clear ordered density, this loop was not modelled due to low homology to available structures and a lack of consensus in secondary structure prediction (see Materials and methods). In spite of this lack of consensus, density in this region suggests that part of this loop may form further α-helical structures. Other TPR-containing α-solenoid proteins form

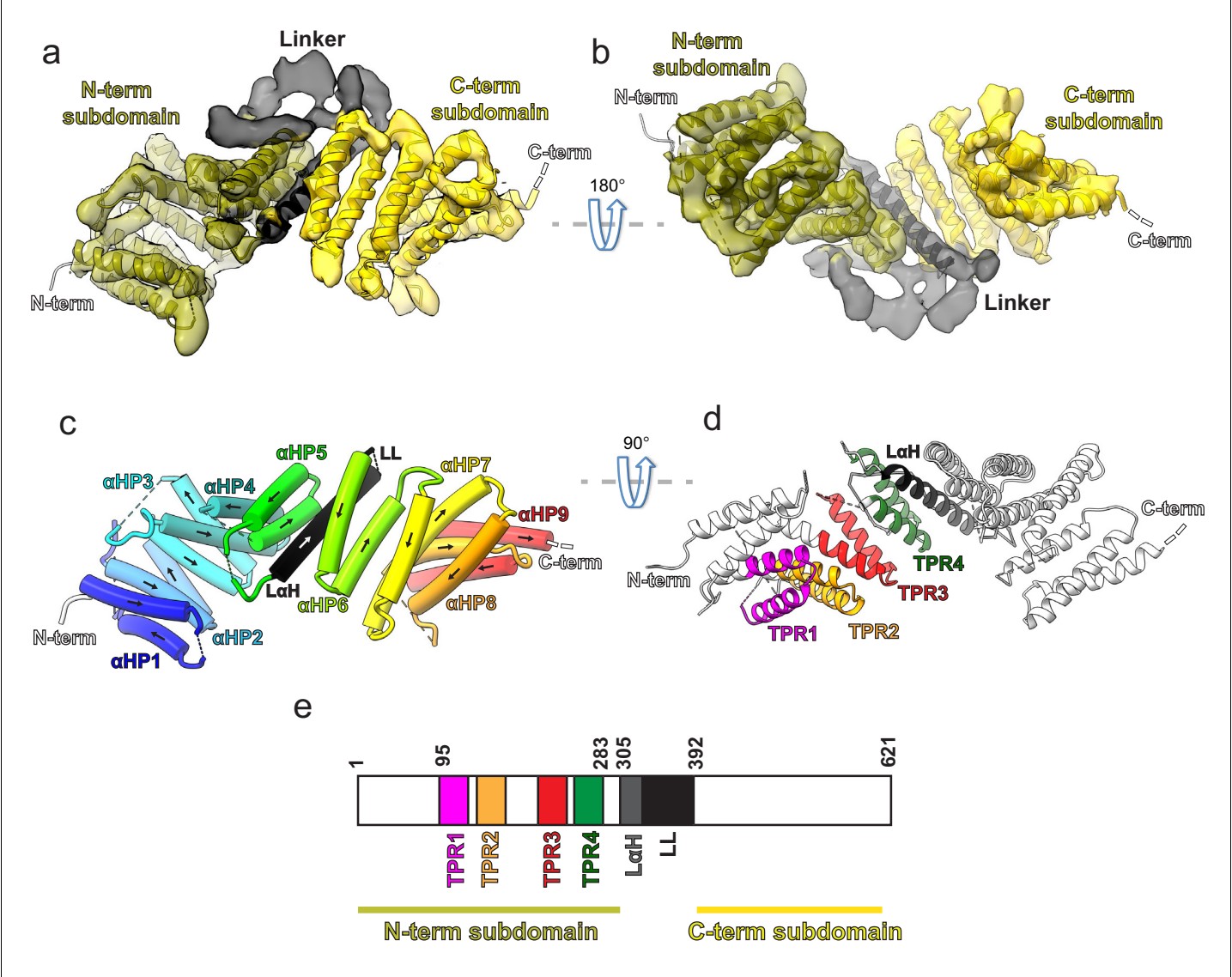

**Figure 1.** Kinesin-binding protein (KBP) is a tetratricopeptide repeat (TPR)-containing right-handed α-solenoid. (**a**) Model of KBP (ribbon representation) displayed in experimental cryo-electron microscopy density. The N-terminal (olive) and C-terminal (gold) subdomains are separated by a linker region (black). Semi-transparent density is coloured regionally as per the fitted model. The N- and C-termini are shown, with a dotted line representing the disordered C-terminus (not modelled). The linker loop (LL) region was not modelled but its density is shown in semi-transparent black. (**b**) The same as panel a, but rotated 180° around the axis indicated. (**c**) The same view as in panel a, but with the density removed and α-helices displayed as pipes with their directionality indicated by arrows. The nine antiparallel α-helical pairs (αHP1–αHP9) are each coloured separately and labelled, as is the linker α-helix (LαH) and LL (dotted line). (**d**) Ribbon representation of KBP showing the four TPR motifs and the LαH coloured according to the labels. View related to panel c, by a 90° rotation around the indicated axis. (**e**) Schematic of the KBP showing the position of the TPR motifs between residue 95 and 283 of the N-terminal subdomain and position of the linker region (LαH and LL) between residues 305 and 392. The online version of this article includes the following figure supplement(s) for figure 1:

**Figure supplement 1.** Kinesin-binding protein (KBP) reconstruction, structure, and loop lengths.

**Figure supplement 2.** Kinesin-binding protein (KBP) loops, sequence, inter-species conservation, and experimental mutations.

**Figure supplement 3.** Approximate path of the kinesin-binding protein (KBP) linker loop (LL).

**Table 1.** Cryo-electron microscopy reconstruction information and model refinement statistics and model geometry.
Data collection, processing, and model refinement information for the kinesin-binding protein (KBP), KBP–KIF15_MD6S, and KIF15_MD6S–MT datasets.

| | KBP (EMDB: EMD-11338, PDB: 6ZPG) | KBP–KIF15_MD6S (EMDB: EMD-11339, PDB: 6ZPH) | KIF15_MD6S–MT (EMDB: EMD-11340, PDB: 6ZPI) |
|---|---|---|---|
| Data collection and processing | | | |
| Pixel size (Å)* | 1.055, 1.043, or 1.047 | 1.047 | 1.39 |
| Number of micrographs (collected, final)* | 9360, 7547 | 6497, 5138 | 214,202 |
| Final particle number | 258,049 (81,628 of which on graphene oxide) | 7513 | 12,674 |
| Map resolution (Å) FSC threshold[†] | 4.6 Independent half-map FSC 0.143 | 6.9 Independent half-map FSC 0.143 | 4.5 Independent half-map FSC 0.143 |
| Refinement | | | |
| Refinement resolution (Å) CC_mask[‡] | 4.6 0.64 | 6.9 0.74 | 6 0.60 |
| Map sharpening $B$-factor (Å$^2$) | −200 | −495 | −134 |
| Model composition Nonhydrogen atoms Protein residues Ligands | 3808 610 0 | 6232 948 1 | 9420 1185 4 |
| R.m.s. deviations[§] Bond lengths (Å) Bond angles (°) | 0.01 0.96 | 0.01 1.07 | 0.08 0.17 |
| Validation[#] MolProbity score Clashscore Poor rotamers (%) | 1.66 5.25 0.5% | 1.84 7.31 0.9% | 1.95 13.25 0.1% |
| Ramachandran plot[#] Favoured (%) Allowed (%) Outliers (%) | 94.38 5.62 0 | 93.13 6.87 0 | 95.38 4.62 0 |

[*]Inclusive of all data collection sessions.

[†]The resolution value at the gold-standard Fourier Shell Correlation (FSC) 0.143 criterion between independently refined half-maps.

[‡]Cross-correlation provided by Phenix real-space refine (*Afonine et al., 2018*).

[§]Root-mean-square deviations of bond lengths or angles in the model.

[#]As defined by the MolProbity validation server (*Chen et al., 2010*).

important regulatory interactions in numerous contexts, and the structure we describe is indicative of similar properties for KBP.

## KBP conformationally adapts to bind KIF15's motor domain using both subdomains

To elucidate the mechanism of kinesin inhibition by KBP, we determined the structure of KBP in complex with the human KIF15 (kinesin-12) motor domain (KIF15_MD, 1–375). This construct, which has six of its eight cysteine residues mutated to serine (C5S, C50S, C162S, C294S, C314S, and C346S) and two additional cysteines were inserted (S250C and G375C), has comparable steady-state ATPase activity to previously published reports (*Klejnot et al., 2014*; *Figure 2—figure supplement 1a*) and we refer to it as KIF15_MD6S. The overall resolution of this KBP–KIF15_MD6S complex was 6.9 Å, with KBP and KIF15_MD6S determined to similar local resolutions (*Figure 2—figure supplement 1b and c*). We built a model of the complex via flexible fitting using our KBP model and the KIF15_MD crystal structure (*Figure 2a and b*, *Table 1* and see Materials and methods). The complex is arranged such that KIF15_MD6S sits in the concave face of the KBP α-solenoid, analogous to

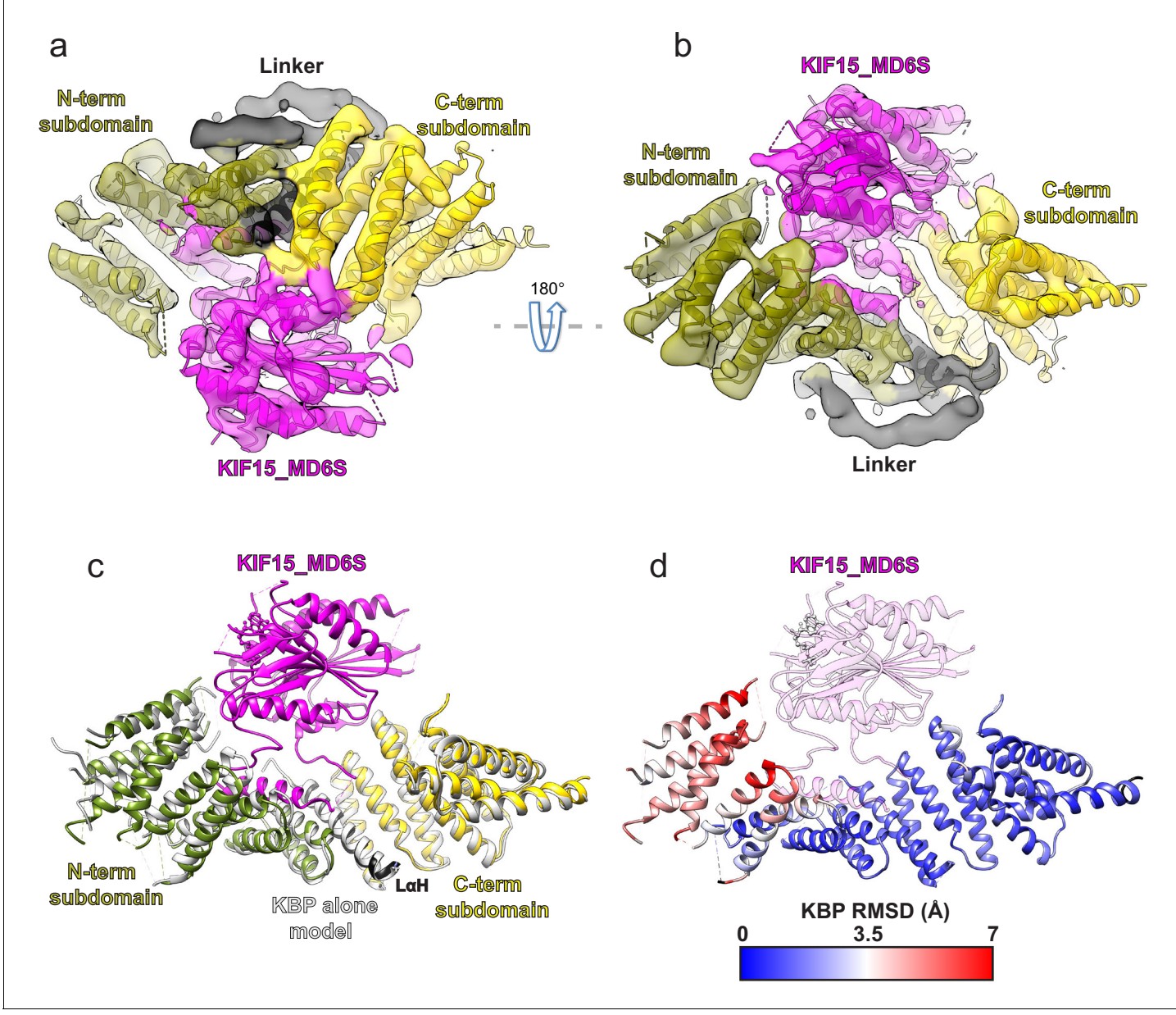

**Figure 2.** Kinesin-binding protein (KBP) conformationally adapts to bind KIF15's motor domain via both subdomains. (a) Model of the KBP–KIF15_MD6S complex (ribbon representation) displayed in experimental cryo-electron microscopy density. The N-terminal (olive) and C-terminal (gold) subdomains and the linker helix (black) are shown in KBP, while kinesin is coloured in magenta. Semi-transparent density is coloured regionally as per the fitted model and additional density for the linker loop is shown in semi-transparent black. (b) The same as panel a, but rotated 180˚ around the axis indicated. (c) The KBP-alone model (light grey ribbons) was superimposed on the KBP–KIF15_MD6S model (opaque ribbons) using Chimera's matchmaker (*Pettersen et al., 2004*). Colouring and view as in panel b. (d) RMSD in Å for KBP comparing KBP–KIF15_MD6S and superimposed KBP-alone models as in panel c, shown on KBP from the KBP–KIF15_MD6S model. Parts of the KBP model coloured black are disordered/missing in the KBP alone model. The KIF15_MD6S is shown in transparent magenta.

The online version of this article includes the following figure supplement(s) for figure 2:

**Figure supplement 1.** Kinesin-binding protein (KBP)–KIF15_MD6S reconstruction resolution estimation and 2D class analysis of KBP–KIF1A_MD and KBP–KIF15_MD complexes.

a baseball enclosed in a baseball glove. The kinesin MD is positioned centrally between the N- and C-terminal subdomains and contacts the KBP concave face and loops at the α-solenoid edges.

When the structure of KBP-alone is superimposed onto KBP in the KBP–KIF15_MD6S complex, it is clear that KBP undergoes a conformational change in the presence of its kinesin motor domain-

binding partner, with the largest differences resulting from an unfurling motion of its N-terminal subdomain (*Figure 2c and d* and *Video 1*). The KBP-alone model is incompatible with KIF15_MD6S binding, due to clashes with L14 in the C-terminal subdomain and αHP3a, αHP4a, and L8 in the N-terminal subdomain. The conformational changes in KBP upon KIF15_MD6S binding relieve these clashes in the complex (*Video 1*).

To establish whether the KBP–KIF15_MD6S mode of interaction applied to other kinesins, we also collected data of the complex formed by KBP with the motor domain of the human kinesin-3 KIF1A (KIF1A_MD). Two-dimensional classification of these images revealed a number of classes with an extra-density corresponding to the size of a kinesin motor domain bound to the concave face of KBP, consistent with what was observed in the KBP–KIF15_MD6S dataset (*Figure 2—figure supplement 1c*). However, in contrast to the KBP–KIF15_MD6S sample, these KBP–KIF1A_MD 2D classes provided only limited views of the complex (*Figure 2—figure supplement 1c and d*), such that a reliable 3D structure could not be calculated. Intriguingly, in addition, the extra kinesin density in the 2D classes appeared to have a somewhat flexible position relative to KBP. However, these data did allow us to confirm that indeed KIF1A_MD also interacts with KBP on its concave face in the same way as KIF15_MD6S and suggests a common mechanism of kinesin inhibition by KBP.

## KIF15_MD6S binds KBP via rearrangement of its tubulin-binding subdomain

We examined the effect of KBP binding on the conformation of KIF15_MD6S. Kinesin motor domains can be structurally divided into three distinct subdomains (*Shang et al., 2014*; *Gigant et al., 2013*) which undergo coordinated conformational changes during the MT-based kinesin ATPase cycle. MT binding stabilises the tubulin-binding subdomain of the MD while the P-loop and Switch 1/2 subdomains – which contain the conserved nucleotide-coordinating P-loop and Switch 1 and 2 motifs – move relative to each other in response to the nucleotide state of the MD (*Shang et al., 2014*; *Gigant et al., 2013*; *Atherton et al., 2014*). We determined the structure of the MT-bound, AMPPNP state of KIF15_MD6S, which shows that this MD adopts a canonical conformation (*Figure 3—figure supplement 1*). Comparison of this conformation with an ADP-bound Kif15_MD crystal structure (PDB: 4BN2 *Klejnot et al., 2014*) illustrates the scale of these MT- and nucleotide-dependent subdomain rearrangements in KIF15, which are similar to those seen in other kinesins MDs (*Shang et al., 2014*; *Atherton et al., 2014*; *Atherton et al., 2017*; *Figure 3—figure supplement 1d and e* and *Figure 3c and d*).

The structure of the KBP–KIF15_MD6S complex revealed that KBP binds the kinesin motor domain via the tubulin-binding subdomain (*Figure 3*). While the P-loop and Switch 1/2 subdomains of the KIF15_MD crystal structure and associated Mg$^{2+}$-ADP generally fitted well into density of the KBP–KIF15_MD6S complex, a large portion of the tubulin-binding subdomain did not (*Figure 3a* and *Figure 3—figure supplement 2a and b*). In particular, there is a striking lack of density in the expected position for helix

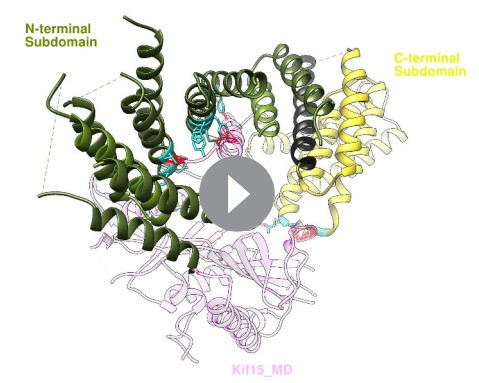

**Video 1.** Kinesin-binding protein (KBP) undergoes conformational change to relieve clashes when forming a complex with KIF15_MD6S. The KBP-alone model was superimposed on the KBP–KIF15_MD6S model using UCSF Chimera's matchmaker (*Pettersen et al., 2004*). A conformational morph movie was then generated in Chimera between the KBP-alone and KIF15 motor domain bound states, with KIF15_MD6S shown throughout to illustrate the relief of clashes. The N-terminal and C-terminal subdomains are coloured in olive and gold respectively, as in *Figure 2a and b*, while KIF15_MD6S is shown in pale magenta. Distances between identified clashing atoms when KBP-alone is superimposed onto the KBP–KIF15_MD6S model are indicated by red linking lines and KBP clashing residues and side chains shown in cyan. Atoms that were clashing remain coloured while the red lines gradually disappear as the clashes are relieved by the conformational change. Clashes were calculated in Chimera using default criteria.
https://elifesciences.org/articles/61481#video1

α4 (*Figure 3a* and *Figure 3—figure supplement 2b*). Instead, there was a strong density of length and width consistent with helix α4 displaced by ~15 Å into the concave face of KBP, which we modelled as such (*Figure 3b and e*, *Figure 2*, and *Figure 3—figure supplement 2b–e*). This displacement of helix α4, which lies close to the TPR-repeat region of the N-terminal subdomain of KBP, is accompanied by additional rearrangements of the flanking L11 and L12 in KIF15_MD6S (labelled KL11 and KL12; *Figure 3c–e* and *Figure 3—figure supplement 2b–e*). A number of other TPR-containing α-solenoids are known to bind peptide motifs with α-helical content within their concave faces (*Culurgioni et al., 2011*; *Wang et al., 2009*; *Quinaud et al., 2007*; *Figure 3—figure supplement 3*), and our structure shows that KBP binds helix α4 of KIF15_MD6S in a similar way.

The KBP-bound conformation of the KIF15_MD6S tubulin-binding subdomain is also radically different from its MT-bound conformation (*Figure 3d and e*). The tubulin-binding subdomain forms the majority of the MT-binding surface in the KIF15_MD6S-MT complex (*Figure 3d* and *Figure 3—figure supplement 1*) such that KBP and MTs cannot simultaneously bind KIF15_MD6S due to extensive steric overlap (*Figure 3d and e*). In summary, KBP sequesters and blocks the MT-interacting surface of kinesin motor domains via a mechanism that involves significant conformational change within the motor domain.

## KBP binds kinesin motor domains via conserved motifs in the α-solenoid edge loops and α-helices at the concave face

KBP contacts the KIF15_MD6S both via (1) loops connecting the α-solenoid edges and (2) TPR-containing α-helices at the concave face (*Figure 4*, *Figure 4—figure supplement 1*, and *Video 2*). At the α-solenoid edges, L1, L3, L5, and L10 in the N-terminal subdomain and L12, L14, L16, and L18 in the C-terminal subdomain are close enough to KIF15_MD6S to be involved in binding. The closest interaction of these was KBP L12 and L14, which contact both Kβ5–KL8 and KL12–Kα5–KL13 regions of the KIF15 tubulin-binding subdomain (*Figure 4*). KBP's disordered L1 lies close to KIF15_MD6S's KL9, while the shorter, ordered L3 and L5 are situated near but not contacting KL11 and Kα6 (*Figure 4—figure supplement 1a and b*). KBP's C-terminal L16 and L18 are close enough to KIF15_MD6S that they may interact with the flexible KL12, N-terminus, or neck-linker. At the TPR-containing region of the concave face of KBP, αHP4a, αHP4b, and αHP5a contact the K11–Kα4–KL12 region of KIF15_MD6S (*Figure 4c and d*).

To test the functional significance of this interface, we investigated KBP–kinesin interactions in cells and examined the activities of mutant KBP constructs in which the predicted interacting amino acids within potentially kinesin-contacting loops were substituted for Ala, Gly, or Pro residues (*Figure 1—figure supplement 2* and Table 2). Ala-substitutions in the TPR-containing α-helices at the KBP concave face were also introduced at particularly inter-species conserved polar residues predicted to interact with the KIF15_MD K11–Kα4–KL12 region (Tyr-213 and Gln-216 in αHP4a, Gln-238 in αHP4b, Thr-255 and Gln-258 in αHP5a; *Figure 4c and d* and *Figure 1—figure supplement 2*). All mutant constructs exhibited roughly equivalent expression patterns that were also comparable to WT KBP (*Figure 5—figure supplement 1*).

We first used pull-down assays. Mouse Kif15 or Kif1A constructs consisting of only the motor domain and the first coiled-coil region (Kif15_MDC or Kif1A_MDC) were fused to bioGFP and co-expressed with various HA-tagged human KBP constructs in HEK293T cells, followed by pull-down of HA–KBP by the bioGFP-KIF_MDC (*Kevenaar et al., 2016*; *Figure 5—figure supplement 2*). Although there are moderate qualitative differences in binding by the two motors, the effects of KBP mutations on motor binding – described in the following – are essentially the same. Ala-substitutions in the TPR-containing α-helices at the KBP concave face (αHP4a and αHP4b), which lie at the heart of the KBP–KIF15_MD6S structural interface, strongly reduced KBP's interaction with both KIF15_MDC and KIF1A_MDC. αHP5a mutants had a similar but less pronounced effect (*Figure 5—figure supplement 2b and c*). In contrast, mutation of L1, L3, or L5 in the KBP N-terminal subdomain or L10 or L16 in the C-terminal subdomain – none of which form directly visualised interactions with KIF15_MD6S in the cryo-EM reconstruction – has no effect on KBP's interaction with either KIF15_MDC or KIF1A_MDC (*Figure 5—figure supplement 2b and c*). Mutation of L12 (to some extent) and of L14 (to a greater extent) – which contact both Kβ5–KL8 and KL12–Kα5–KL13 – reduced KBP interaction with KIF15_MDC and KIF1A_MDC (*Figure 5—figure supplement 2b and c*). Mutation of L12 + L14 additively disrupted the KBP–motor interaction, consistent with the structural proximity of these two loops in the kinesin–KBP complex. L10 + L12 and L10 + L14

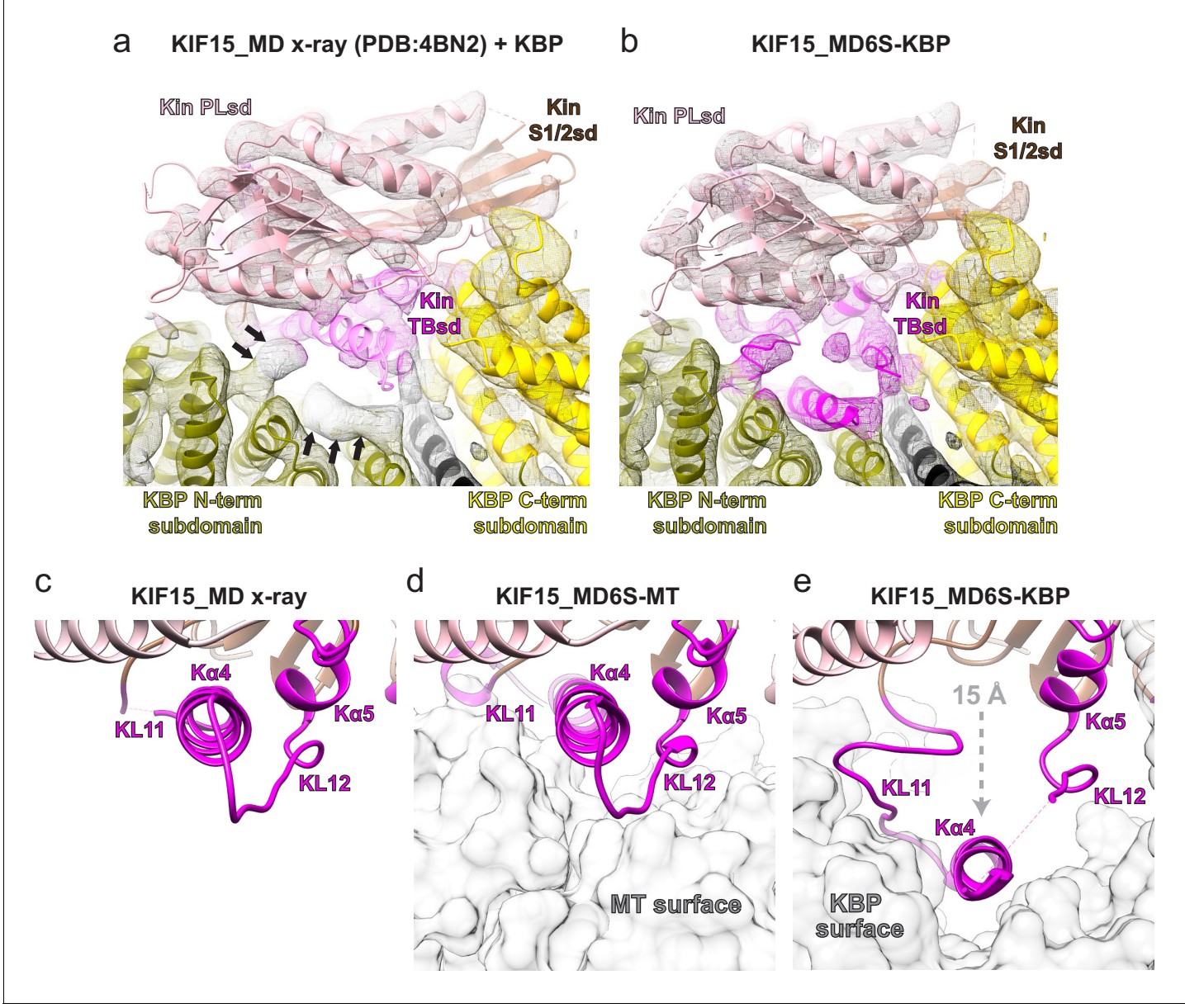

**Figure 3.** The KIF15 motor domain binds kinesin-binding protein (KBP) via rearrangement of its tubulin-binding subdomain. (a) The crystallographic model of the KIF15_MD alone (PDB: 4BN2 *Klejnot et al., 2014*) was superimposed on the KIF15 region of the KBP–KIF15_MD6S complex, with the KIF15 part of the KBP–KIF15_MD6S complex model hidden. The KIF15_MD6S Switch 1/2 subdomain (Switch 1/2 subdomain) is coloured sienna, and the P-loop subdomain (Kin-PLsd) is coloured light pink. The TBsd of the KIF15_MD crystallographic model is shown as pale magenta to illustrate poor fit into density. The KBP subdomains are coloured as labelled. Black arrows indicate unaccounted-for cryo-electron microscopy (cryo-EM) density. Individual secondary structure elements in the tubulin-binding subdomain are labelled. The cryo-EM density for the KBP–KIF15_MD6S complex is shown in mesh and is coloured by proximity (≤3.5 Å) to the fitted model. (b) Same as in panel a, but the whole fitted KBP–KIF15_MD6S complex model is shown. The KIF15_MD6S tubulin-binding subdomain (TBsd) is now coloured magenta to indicate good fit into density. (c) Zoomed view of just the TBsd (corresponding to the boxed region in *Figure 3—figure supplement 2d*), showing just the KIF15_MD-alone crystallographic model. (d) The TBsd in the KIF15_MD6S-MT model, same view as in panel c. The MT is shown in light grey surface representation. (e) The TBsd in the KBP–KIF15_MD6S model, same view as in panel c. KBP is shown in light grey surface representation and the ~15 Å displacement of helix α4 is indicated by the dashed grey arrow.

The online version of this article includes the following figure supplement(s) for figure 3:

**Figure supplement 1.** KIF15_MD6S adopts a canonical MT-bound kinesin conformation.

**Figure supplement 2.** Movement of Kα4 of the Kin TBsd upon kinesin-binding protein (KBP) binding.

**Figure supplement 3.** Examples of tetratricopeptide repeat (TPR)-containing α-solenoid proteins binding α-helical SSE ligands.

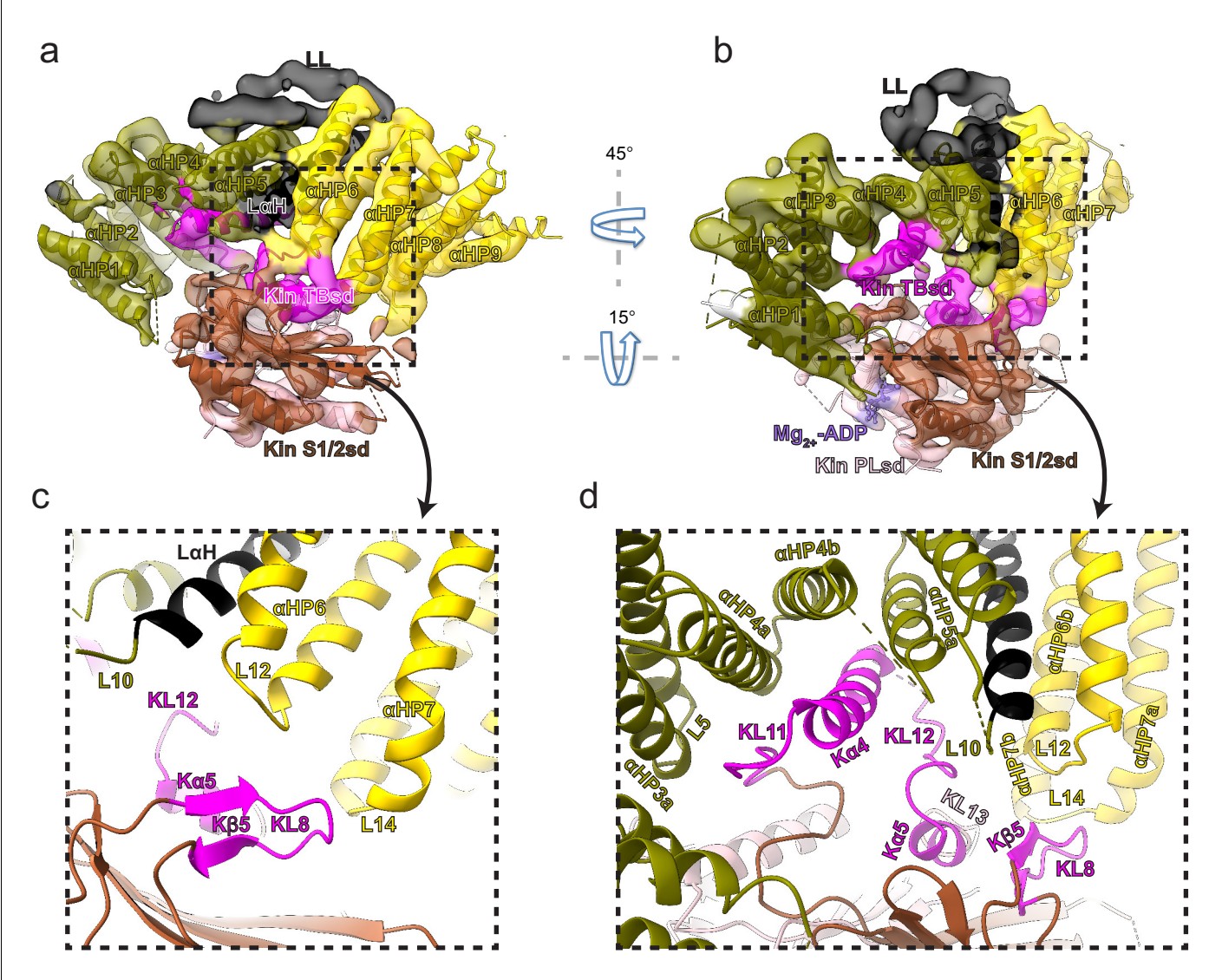

**Figure 4.** Kinesin-binding protein (KBP) binds kinesin MDs via conserved motifs in the α-solenoid edge loops and α-helices at the concave face. (**a**) Pseudo-atomic model of the KBP–KIF15_MD6S complex (ribbon representation) displayed in cryo-electron microscopy density, using the same viewpoint as *Figure 2a*, but with the KIF15_MD6S now coloured by subdomain as in *Figure 3*. The KIF15_MD6S Switch 1/2 subdomain (Kin S1/2 sd) is coloured sienna, and the P-loop subdomain (Kin-PLsd) is coloured light pink. The KIF15_MD6S tubulin-binding subdomain (TBsd) is coloured magenta. The KBP subdomains are coloured as labelled. The nine helix pairs of KBP are labelled. Semi-transparent density is coloured regionally as per the fitted model and additional density for the linker loop is shown in semi-transparent black. (**b**) The same as panel a, but rotated 45° and 15° respectively around the axes indicated. (**c**) Zoomed view of the region indicated in panel a, with density removed and selected KIF15_MD6S and KBP secondary structure elements labelled. (**d**) Zoomed view of the region indicated in panel b, with density removed and selected KIF15_MD6S and KBP secondary structure elements labelled.

The online version of this article includes the following figure supplement(s) for figure 4:

**Figure supplement 1.** Additional kinesin-binding protein (KBP) α-solenoid edge loops proximal to KIF15_MD6S.

mutants also had weaker interactions with KIF15_MDC/KIF1A_MDC (*Figure 5—figure supplement 2b and c*), again pointing to the additive contributions of loops in the KBP C-terminal subdomain to kinesin binding. Strikingly, mutation of L18 appears to enhance the interaction between KBP and both KIF15_MDC and KIF1A_MDC, suggesting that it may somehow contribute to negative regulation of binding in the context of WT KBP.

We then used a previously described inducible peroxisome translocation assay in COS-7 cells (*Kevenaar et al., 2016*). In this assay, dimeric mouse Kif15_MDC or Kif1A_MDC constructs with an

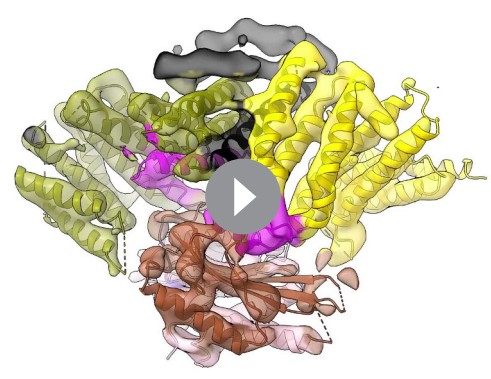

**Video 2.** Interaction of kinesin-binding protein (KBP) with the KIF15 motor domain. Model of the KBP–KIF15_MD6S complex (ribbon representation) displayed in experimental cryo-electron microscopy density. The N-terminal (olive) and C-terminal (gold) subdomains and the linker region (black) are shown in KBP, while the KIF15_MD6S Switch 1/2 subdomain (Switch 1/2 subdomain) is coloured sienna, the P-loop subdomain (Kin-PLsd) is coloured light pink and the Kif15_MD tubulin-binding subdomain (TBsd) is coloured magenta. Semi-transparent density is coloured regionally as per the fitted model and additional density for the linker loop is shown in semi-transparent black.

https://elifesciences.org/articles/61481#video2

FRB-tag (Kif15_MDC–FRB or Kif1A_MDC–FRB) are expressed together with PEX–mRFP–FKBP, a peroxisome-binding construct, along with the various KBP constructs. Addition of rapalog induces FRB–FKBP heterodimerisation and motor-driven peroxisome translocation to the cell periphery, but when the motor is inhibited by KBP, peroxisome translocation is blocked (*Figure 5a–c*). Kinesin-mediated translocation was measured, first by quantifying the number of cells in which peroxisome translocation is seen (*Figure 5d and e*), and second, by quantifying peroxisome intensities above a threshold value in the cell periphery (*Figure 5f and g*). Because of observed differences in peroxisome translocation within the time-frame of rapalog treatment, different peroxisome intensity threshold values and peripheral areas were used for KIF1A and KIF15; this is probably due to differences in motor properties (*Figure 5—figure supplement 3*).

Intriguingly, while the overall trends in perturbation of KBP inhibition by mutagenesis seen in the pull-down assay are recapitulated in the translocation assay, some differences are also observed. As with the pull-down assay, Ala-substitutions in the TPR-containing α-helices αHP4a and αHP4b at the KBP concave face, as well as αHP5a, all strongly reduced KBP's inhibition of both KIF15_MDC and KIF1A_MDC peroxisome translocation activity (*Figure 5d–g*). This can be seen by the extent of peroxisome translocation and in an increase of peroxisomes in the cell periphery after rapalog addition, similar to the control condition without KBP (*Figure 5d–g*).

Also, as observed in the pull-down assay, mutation of L1, L3, or L5 had no effect on KBP's inhibition of KIF15_MDC or KIF1A_MDC (*Figure 5d–g*). This reinforces the conclusion that while these elements are close enough to form contacts with the parts of the kinesin motor domain in our reconstruction, they do not contribute significantly to KBP inhibition of kinesin-mediated translocation in cells.

Mutation of either of L12 or L14 strongly abrogated KBP inhibition of KIF15_MDC/KIF1A_MDC-based translocation, a more pronounced effect than was seen in the pull-down assay. Similarly, although mutation of L10 and L16 had no effect on KBP–kinesin interaction in the pull-down assay, mutation of these loops disrupted KBP inhibition of KIF1A_MDC and KIF15_MDC in the translocation assay (*Figure 5d–g*). A subset of the above described mutations was also combined to assess additive effects. Here we observed that KBP constructs containing mutations in both L10 + L12, L10 + L14, or L12 + L14 had similar effects to KBP with only one of the loops mutated (*Figure 5d–g*), suggesting that KBP inhibition is more readily disrupted in the translocation assay.

Interestingly, while all the above described regions affected KBP inhibition of KIF15_MDC and KIF1A_MDC equivalently, the L18 KBP mutant inhibited KIF15_MDC equivalent to wild type, but only exhibited partial inhibition of KIF1A_MDC-mediated translocation. It should, however, be noted that fewer cells show peroxisome translocation by KIF1A_MDC when L18 is co-expressed (*Figure 5e*), suggesting that the L18 KBP mutant also inhibits KIF1A_MDC to some extent. The L18 mutant is the single example of contradictory behaviours between the assays, because it appeared to enhance KBP–kinesin binding in the pull-down assay. Our structural data do not provide a clear rationale for this, and future studies will investigate the role of this region of KBP further and, for

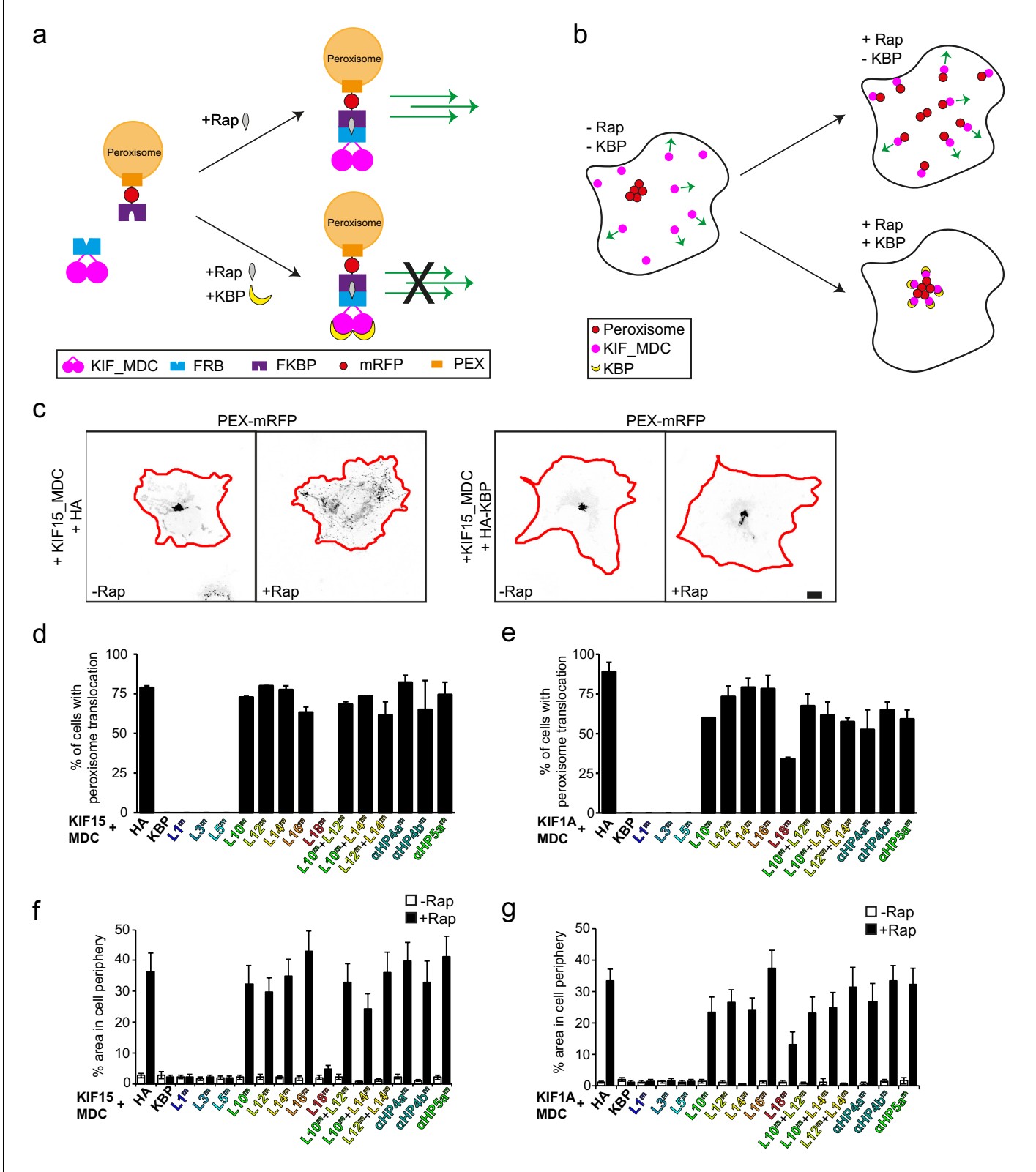

**Figure 5.** Disruption of cryo-electron microscopy defined kinesin-binding protein (KBP)–kinesin interface perturbs KBP inhibition of KIF15- and KIF1A-mediated cargo translocation in cells. (a) Schematic depiction of the inducible peroxisome motility assay, with the kinesin motor domain fused to an FRB domain and PEX fused to an FKBP domain. Addition of rapalog (Rap) links FRB and FKBP and induces peroxisome translocation by kinesin dimers. Expression of KBP inhibits kinesin movement, such that addition of rapalog cannot induce peroxisome translocation. (b) Schematic representation of
*Figure 5 continued on next page*

*Figure 5 continued*

the inducible peroxisome motility assay in cells. Without rapalog or KBP, peroxisomes localise in the cell centre, whereas kinesin moves towards the cell periphery. Rapalog induces peroxisome translocation into the cell periphery, which is inhibited in the presence of KBP. (c) Representative images of peroxisomes in COS-7 cells expressing KIF15_MDC–FRB, PEX–mRFP–FKBP, and HA (left panels) or HA–KBP (right panels) without and with addition of rapalog. Scale bar, 10 µm. (d, e) Quantification of the percentage of cells in which peroxisome translocation is observed after rapalog treatment in cells expressing KIF15_MDC–FRB (d) or KIF1A_MDC–FRB (e), PEX–mRFP–FKBP, and HA–KBP constructs including the indicated mutants. Data are displayed as mean ± s.e.m. (n = 28–35 cells from two independent experiments). (f, g) Quantification of the area above threshold intensity in the outer 5 µm (KIF1A_MDC) or 7.5 µm (KIF15_MDC) of the cell from the total area above threshold intensity in cells expressing KIF15_MDC–FRB (f) or KIF1A_MDC–FRB (g), PEX–mRFP–FKBP, and HA–KBP constructs including the indicated mutants without and with rapalog treatment. Data are displayed as mean ± s.e.m. (n = 28–35 cells from two independent experiments).

The online version of this article includes the following figure supplement(s) for figure 5:

**Figure supplement 1.** Kinesin-binding protein (KBP) mutants show similar expression profiles in COS-7 cells.

**Figure supplement 2.** Pull-down experiments demonstrate the effect of kinesin-binding protein (KBP) mutation on the interaction between KIF15 and KIF1A.

**Figure supplement 3.** Kinesin motors show different properties in the peroxisome assay.

example, whether it is subject to post-translational regulation that could regulate KBP's inhibitory activity.

However, taken together, the translocation and pull-down assays both demonstrate the functional importance of the kinesin interaction with the TPR-containing α-helices at the KBP concave face and the set of loops in the KBP C-terminal subdomain. The translocation and pull-down assays also reveal differences in the sensitivity of the KBP–kinesin interaction to perturbation, where translocation is more readily disrupted than the interactions detected by pull-down. These differences likely reflect the greater complexity of motor regulation during active translocation and could be a function of FRB–FKBP-mediated motor dimerisation. It might also reflect the fact that cellular MTs in the translocation assay can directly compete with KBP for kinesin binding. Overall, these mutation studies support the idea that KBP interacts with different kinesin family members in a similar way via an extended interface at KBP's concave face that is composed of TPR-containing α-helices and α-solenoid edge loops, particularly in the C-terminal subdomain.

## Specific sequences in the tubulin-binding subdomain are conserved across KBP-binding kinesin family members

Given that KBP selectively binds and inhibits only a subset of kinesins (*Kevenaar et al., 2016*), we used our structural data to investigate the basis of this selectivity. Although the resolution of our reconstruction and the flexibility of some loops do not provide a detailed molecular description of the interaction interface, our structure shows that the kinesin tubulin-binding subdomain is the key KBP-interacting region. Analysis of the sequences of this region in KBP-binding and KBP-non-binding kinesins (*Figure 6a*) revealed patterns of sequence conservation across the entire subdomain in all KBP-binding kinesins; this included both the Kβ5–KL8 and KL11–Kα4–KL12–Kα5–KL13 regions. In contrast, the equivalent regions are more variable in kinesins that are not inhibited by KBP. The length of KL8, which joins the two Kβ5 strands, was also consistently five residues long in KBP-binding kinesins, while it was variable in KBP-non-binding kinesins. From our KBP–KIF15_MD6S structure, the sensitivity of the KBP interaction to KL8 length makes sense considering the tight fit of this loop between KBP L12 and L14 (*Figure 6b*). In summary, two consensus motifs in KL11–Kα4–KL12–Kα5–KL13 and Kβ5–KL8 regions of the tubulin-binding subdomain are found in KBP-binding kinesins and these are likely to form the basis of KBP's kinesin family member selectivity. We therefore propose a model where KBP selects and inhibits target kinesins through binding and remodelling a compatible tubulin-binding subdomain, obstructing the kinesin MT-binding surface (*Figure 6c*).

## Discussion

In this study, we reveal the TPR-containing right-handed α-solenoid structure of the ~72 kDa KBP using VPP cryo-EM. At the time of writing and to our knowledge, structures of only a few macromolecular complexes <80 kDa have been determined using cryo-EM (*Merk et al., 2016*; *Zhang et al., 2018*; *Khoshouei et al., 2017*; *Herzik et al., 2019*; *Fan et al., 2019*). The structure of the KBP–

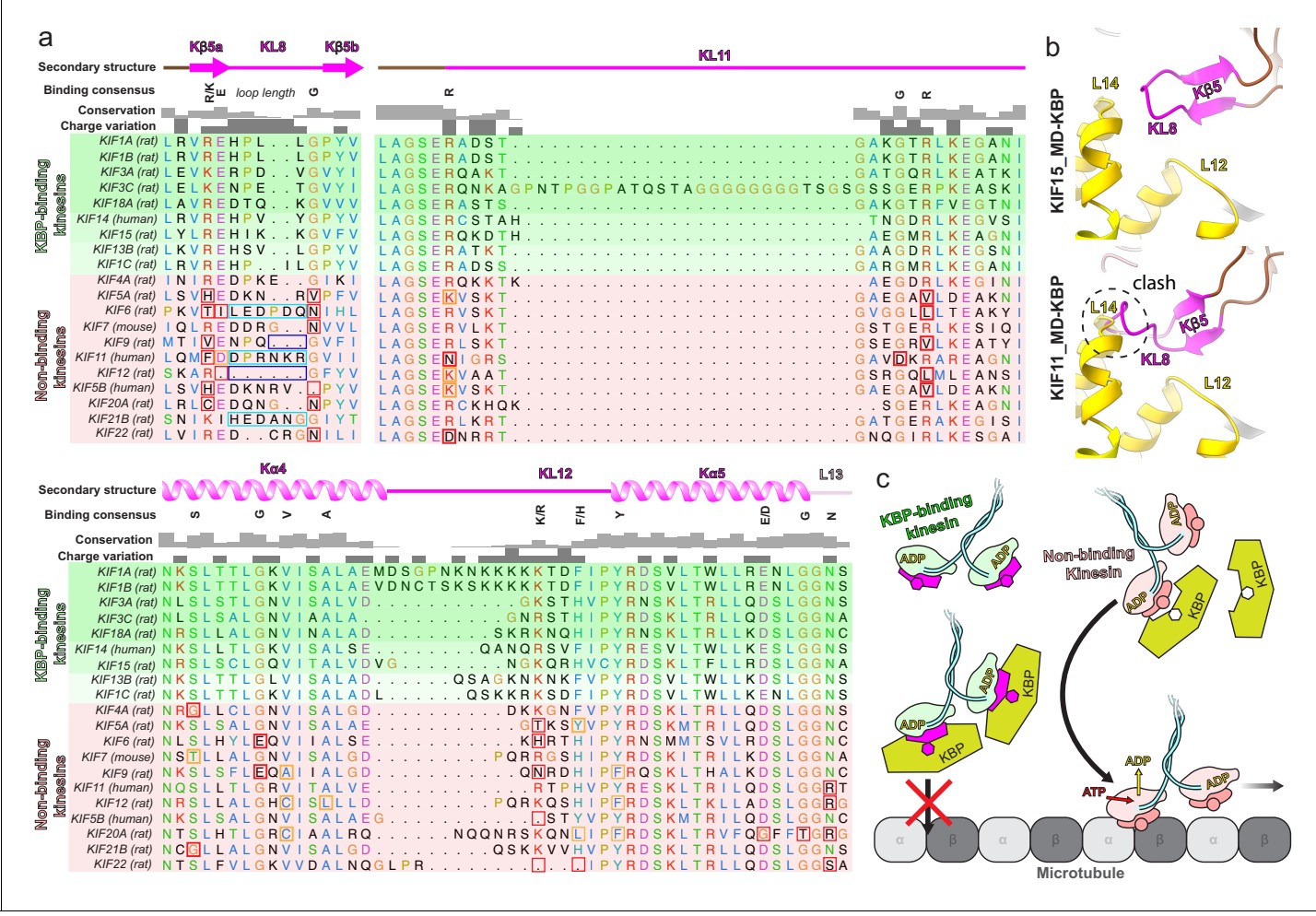

**Figure 6.** Conserved motifs in kinesin-binding protein (KBP)-binding kinesin MDs. (**a**) Sequence alignment of the tubulin-binding subdomain from kinesin motor domains, made using Clustal Omega multiple sequence alignment (*Sievers et al., 2011*). Residues are coloured according to standard Clustal X colouring (dependent on residue type and conservation, see http://bioinfolab.unl.edu/emlab/documents/clustalx_doc/clustalx.html#C). Kinesin MD constructs experimentally assessed for KBP interactivity are taken from *Kevenaar et al., 2016*; strongest interactors are in rows highlighted in darker shades of green, weaker interactors in lighter shades of green, and non-interactors in red. Secondary structure element, conservation and charge variation columns, as well as a 'binding consensus' column indicating residues/loop length conserved at the interface (according to the KBP–KIF15_MD6S complex) in KBP-binding but not non-binding kinesins are shown above the alignment. Non-conservation relative to this consensus is shown in boxed sequence; red boxes, non-conservative substitutions, orange boxes, conservative substitutions (general charge/polarity/hydrophobicity retention), cyan boxes, extended loop region, dark blue boxes, truncated loop region. (**b**) Top panel; view of the Kβ4–KL8 region of the tubulin-binding subdomain in the KBP–KIF15_MD6S model, coloured as in *Figure 4*. Bottom panel; as in upper panel, but with the KIF11_MD cryo-electron microscopy model (PDB: 6TA4 [*Peña et al., 2020*]) superimposed onto the now hidden KBP–KIF15_MD6S. Note steric clash introduced by KIF11_MD's extended KL8. (**c**) Schematic model of KBP's hypothesised selective kinesin inhibition mechanism. KBP (olive) binds the compatible TBsd of recognised kinesins (magenta) but is incompatible with the TBsd of non-binding kinesins (salmon). For its target kinesins, KBP therefore sterically blocks the TBsd interaction with MTs (grey), preventing activation of kinesin ATPase and motility.

KIF15_MD6S complex shows how KBP binds the KIF15_MD6S via its concave face and undergoes subtle remodelling of its N-terminal domain to accommodate kinesin binding. This further reinforces the idea that the TPR-containing structures are not simply static scaffolds but can flexibly respond to ligand binding (*Pernigo et al., 2018*). In contrast, the KIF15 motor domain undergoes a radical conformational change in forming a complex with KBP, in which helix α4, the major component of the motor's tubulin-binding subdomain, is displaced from the main MD body by ~15 Å into the KBP concave face. This is consistent with previous observations that this region of the kinesin motor domain being rather malleable and able to move independently of the core structure of kinesin motor domains (*Wang et al., 2016a*; *Scarabelli and Grant, 2013*). Our evidence suggests this observation

is not due to use of a cysteine-substituted KIF15 motor domain construct because: (i) this protein exhibited equivalent MT-stimulated ATPase activity compared to non-substituted KIF15_MD (*Figure 2—figure supplement 1a*), (ii) it exhibited structurally canonical MT binding and response to nucleotide (*Figure 3—figure supplement 1*), and (iii) the substituted residues are not well conserved among KBP-binding kinesins (*Figure 6a*). The large displacement of helix α4 expands the surface area over which the normally compact tubulin-binding subdomain of the kinesin motor domain can interact with KBP, and it is through the sequence and shape of this interface that the selectivity of KBP for a subset of kinesin motors is presumably defined.

Analysis of the KBP–KIF1A_MD complex (*Figure 2—figure supplement 1c*) supports the idea of a conserved mode of interaction between a subset of kinesins and the concave face of KBP. Interestingly, KIF1A_MD exhibited flexibility in its interaction with KBP, which was not observed in the KBP–KIF15_MD6S complex. Whether this reflects a physiological reality, or a result of the EM preparation method is uncertain at present, although we think it unlikely to be due to our use of KIF15_MD6S. However, our 2D classifications combined with mutation studies strongly suggest KIF15_MD6S and KIF1A_MD share an overall similar KBP-binding mode.

Targeted mutations to the various kinesin-binding KBP elements reduced complex affinity, yet no single mutation completely disrupted the interaction in our pull-down assays (*Figure 5—figure supplement 2*). In contrast, in the context of active MT-based cargo translocation in the cellular environment, KBP inhibitory activity appeared more sensitive to disruption (*Figure 5d–g*). Although most KBP mutations that have been reported in GOSHS result in total loss of protein (*Brooks et al., 2005*; *Dafsari et al., 2015*; *Valence et al., 2013*; *Salehpour et al., 2017*), a recent study details missense mutations that only partially reduce protein expression (*MacKenzie et al., 2020*). Our data illustrate that KBP's activity is additionally sensitive to mutations in key regions that affect its ability to bind kinesins.

Our structure shows that KBP binds exclusively to the tubulin-binding subdomain of KIF15_MD6S, sterically preventing MT attachment. The interaction of kinesin motor domains with the MT surface via its tubulin-binding subdomain stimulates nucleotide exchange and kinesin ATPase activity. In contrast, on interaction with KBP, the displacement of helix α4 from the kinesin nucleotide-binding site, together with the absence of MT-mediated ordering of KL9 and KL11, means that its catalytic site is distorted and the structural changes associated with MT-stimulated ATPase cannot occur. This could be an important facet of the role of KBP in the energy economy of the cell in addition to directly blocking kinesin–MT interactions.

The concave face of TPR-containing α-solenoids commonly serve as a recognition platform for specific peptide motifs, including those forming α-helical structures (*Perez-Riba and Itzhaki, 2019*). Specificity and affinity for target motifs are determined in part by the shape of the α-solenoid concave face, which in turn is defined by the fold's supertwist. In addition, particular amino acid arrangements at the concave face contribute to partner-binding affinity and specificity, together with additional interfaces formed at the convex surface or α-solenoid edge (*Zeytuni and Zarivach, 2012*). KBP kinesin specificity and affinity are defined by the interaction of its concave face with the large surface area of the kinesin L11–Kα4–KL12–Kα5–KL13 region, in addition to binding the kinesin Kβ5–KL8 region at its α-solenoid edge. Interestingly, distal to the N-terminal MT-binding region of kinesin-1, C-terminally associated kinesin light chain use their unique TPR-containing α-solenoid concave face to select cargos via recognition of specific peptide motifs (*Cross and Dodding, 2019*). Therefore, peptide selectivity by TPR-containing α-solenoids is a facet of both kinesin MT-binding and cargo-binding regulatory mechanisms. Such protein–protein interactions may be selectively targeted for disruption (*Randall et al., 2017*), and the insights arising from our work provide future avenues to disrupt KBP–kinesin interactions and thereby explore KBP interactions and regulatory roles.

The effective and selective kinesin inhibitory mechanism of KBP revealed by our work may fulfil specific roles in the kinesin regulatory toolbox employed by cells to spatially and temporarily orchestrate kinesin activity. Future studies will be aimed at understanding how KBP interacts with kinesins in dimeric and/or autoinhibited forms. For example, while KBP does not interact with some target kinesins in autoinhibited conformations (*Malaby et al., 2019*), others that retain a structurally available tubulin-binding subdomain, such as autoinhibited kinesin-3 monomers (*Ren et al., 2018*), are feasible binding partners. KBP can bind to constitutively active dimeric kinesin constructs that lack autoinhibitory regions (*Kevenaar et al., 2016*; *Malaby et al., 2019*), although the stoichiometry and structural details in this context are unclear. Furthermore, any additional effects of kinesin cargo

binding on their susceptibility to KBP inhibition are not well understood. It will also be of key importance to elucidate the mechanisms of KBP activity regulation, for example, by phosphorylation of KBP and/or kinesins (*Kevenaar et al., 2016*) and KBP acetylation and targeted degradation by the ubiquitin system (*Donato et al., 2017*). Our structural characterisation of the KBP–kinesin inhibitory interaction provides an important mechanistic platform from which to expand our understanding of KBP's biological roles in neuronal function and cancer.

# Materials and methods

**Key resources table**

| Reagent type (species) or resource | Designation | Source or reference | Identifiers | Additional information |
|---|---|---|---|---|
| Gene (*Homo sapiens*) | KIAA1279 | GenBank | HGNC:23419 | |
| Gene (*Mus musculus*) | KIF1A | GenBank | MGI:108391 | |
| Gene (*Mus musculus*) | KIF15 | GenBank | MGI:1098258 | |
| Strain, strain background (*Escherichia coli*) | BL21(DE3) | NEB | Cat. #: C2527H | Competent cells |
| Strain, strain background (*Escherichia coli*) | BL21-Gold (DE3) | Agilent | Cat. #: 230130 | Competent cells |
| Strain, strain background (*Escherichia coli*) | Rosetta2 (DE3) | Novagen | Cat. #: 71400 | Competent cells |
| Cell line (*Homo sapiens*) | Human embryonic kidney 239T (HEK293T) | ATCC | CRL-3216 RRID:CVCL_0063 | |
| Cell line (*Cercopithecus aethiops*) | *Cercopithecus aethiops* kidney (COS-7) | ATCC | CRL-1651 RRID:CVCL_0224 | |
| Peptide, recombinant protein | Porcine tubulin (>99% pure) | Cytoskeleton Inc | Cat. #: T240 | |
| Antibody | Anti-HA (mouse monoclonal) | Roche | Cat# 11666606001; RRID:AB_514506 | IF (1:500) |
| Antibody | Anti-mouse IgG1, Alexa488 (goat polyclonal) | Thermo Fisher Scientific | Cat# A-21121, RRID:AB_2535764 | IF (1:400) |
| Antibody | Anti-HA (mouse monoclonal) | Biolegend | Cat# 901533; RRID:AB_2801249 | WB (1:2000) |
| Antibody | Anti-GFP (rabbit polyclonal) | Abcam | Cat# ab290; RRID:AB_303395 | WB (1:10000) |
| Antibody | Anti-rabbit IgG antibody, IRDye 680LT conjugated (goat polyclonal) | LI-COR Biosciences | Cat# 827–11081; RRID:AB_10795015 | WB (1:20000) |

*Continued on next page*

Continued

| Reagent type (species) or resource | Designation | Source or reference | Identifiers | Additional information |
|---|---|---|---|---|
| Antibody | Anti-mouse IgG antibody, IRDye 800CW conjugated (goat polyclonal) | LI-COR Biosciences | Cat# 827–08364; RRID:AB_10793856 | WB (1:15000) |
| Recombinant DNA reagent | KBP (plasmid) | *Kevenaar et al., 2016* | | Described in Materials and methods |
| Recombinant DNA reagent | KIF1A_MD (plasmid) | *Atherton et al., 2014* | | Described in Materials and methods |
| Recombinant DNA reagent | KIF15_MD (plasmid) | This study | | Described in Materials and methods |
| Recombinant DNA reagent | pebioGFP (plasmid) | *van der Vaart et al., 2013* | N/A | Described in Materials and methods |
| Recombinant DNA reagent | BirA coding vector (plasmid) | *van der Vaart et al., 2013* | N/A | Described in Materials and methods |
| Recombinant DNA reagent | GW1–PEX3–mRFP–FKBP1 (plasmid) | *Kevenaar et al., 2016* | N/A | Described in Materials and methods |
| Recombinant DNA reagent | β-actin–Kif1A_MDC–FRB (plasmid) | *Kevenaar et al., 2016* | N/A | Described in Materials and methods |
| Recombinant DNA reagent | β-actin–Kif15_MDC–FRB (plasmid) | This study | N/A | Described in Materials and methods |
| Recombinant DNA reagent | pebioGFP-Kif1A_MDC (plasmid) | This study | N/A | Described in Materials and methods |
| Recombinant DNA reagent | pebioGFP-Kif15_MDC (plasmid) | This study | N/A | Described in Materials and methods |
| Recombinant DNA reagent | pGW1–HA–KBP (plasmid) | This study | N/A | Described in Materials and methods |
| Recombinant DNA reagent | pGW1–HA–KBP_L1 (plasmid) | This study | N/A | Described in Materials and methods |
| Recombinant DNA reagent | pGW1–HA–KBP_L3 (plasmid) | This study | N/A | Described in Materials and methods |
| Recombinant DNA reagent | pGW1–HA–KBP_L5 (plasmid) | This study | N/A | Described in Materials and methods |
| Recombinant DNA reagent | pGW1–HA–KBP_L10 (plasmid) | This study | N/A | Described in Materials and methods |
| Recombinant DNA reagent | pGW1–HA–KBP_L12 (plasmid) | This study | N/A | Described in Materials and methods |
| Recombinant DNA reagent | pGW1–HA–KBP_L14 (plasmid) | This study | N/A | Described in Materials and methods |

*Continued on next page*

*Continued*

| Reagent type (species) or resource | Designation | Source or reference | Identifiers | Additional information |
|---|---|---|---|---|
| Recombinant DNA reagent | pGW1–HA–KBP_L16 (plasmid) | This study | N/A | Described in Materials and methods |
| Recombinant DNA reagent | pGW1–HA–KBP_L18 (plasmid) | This study | N/A | Described in Materials and methods |
| Recombinant DNA reagent | pGW1–HA–KBP_L10+L12 (plasmid) | This study | N/A | Described in Materials and methods |
| Recombinant DNA reagent | pGW1–HA–KBP_L10+L14 (plasmid) | This study | N/A | Described in Materials and methods |
| Recombinant DNA reagent | pGW1–HA–KBP_L12+L14 (plasmid) | This study | N/A | Described in Materials and methods |
| Recombinant DNA reagent | pGW1–HA–KBP_αHP4a (plasmid) | This study | N/A | Described in Materials and methods |
| Recombinant DNA reagent | pGW1–HA–KBP_αHP4b (plasmid) | This study | N/A | Described in Materials and methods |
| Recombinant DNA reagent | pGW1–HA–KBP_αHP5a (plasmid) | This study | N/A | Described in Materials and methods |
| Sequence-based reagent | KBP_fwd | This study | PCR primer for KBP mutants | TATTATTATGGCGCGCCAGGATCCCCGGAATTCGGCACGAGGGAGGCCGCTATGGCGAACGTTCCGTGGGCA |
| Sequence-based reagent | KBP_rev | This study | PCR primer for KBP mutants | CTCGTCGACTCCTAATCCTTAAGTCAGGGCCATCTT |
| Sequence-based reagent | KBP_L1_fwd | This study | PCR primer for KBP_L1 | CTGCATAAAAATCCGGCAGCAGCACCAGCAGCATCCAAATACAGCGCC |
| Sequence-based reagent | KBP_L1_rev | This study | PCR primer for KBP_L1 | GGCGCTGTATTTGGATGCTGCTGGTGCTGCTGCCGGATTTTTATGCAG |
| Sequence-based reagent | KBP_L3_fwd | This study | PCR primer for KBP_L3 | TGAACCACATCGACGCAGGAGGACTGTCGGCGGGGGA |
| Sequence-based reagent | KBP_L3_rev | This study | PCR primer for KBP_L3 | TCCCCCGCCGACAGTCCTCCTGCGTCGATGTGGTTCA |
| Sequence-based reagent | KBP_L5_fwd | This study | PCR primer for KBP_L5 | ATCTTGTGGTCTGAAGCAGGAGCAATTGAAACTGCACAG |
| Sequence-based reagent | KBP_L5_rev | This study | PCR primer for KBP_L5 | CTGTGCAGTTTCAATTGCTCCTGCTTCAGACCACAAGAT |
| Sequence-based reagent | KBP_L10_fwd | This study | PCR primer for KBP_L10 | TTTGGTCAAACTGGAGCAGGAGCAGGAGCAGGAGCAGGACCAGCAGGAGCAGGAGCAGGACCAGGAGGATATCATCAAAGAAA |

*Continued on next page*

*Continued*

| Reagent type (species) or resource | Designation | Source or reference | Identifiers | Additional information |
|---|---|---|---|---|
| Sequence-based reagent | KBP_L10_rev | This study | PCR primer for KBP_L10 | TTTCTTTGATGATATCC TCCTGGTCCTGCTCCT GCTCCTGCTGGTCCT GCTCCTGCTCCTGCT CCTGCTCCAGTTTGACCAAA |
| Sequence-based reagent | KBP_L12_fwd | This study | PCR primer for KBP_L12 | GAGTTCTTTCAGATT GGCGGCGCGGTC ACTGACCATATT |
| Sequence-based reagent | KBP_L12_rev | This study | PCR primer for KBP_L12 | AATATGGTCAGTGA CCGCGCCGCCAA TCTGAAAGAACTC |
| Sequence-based reagent | KBP_L14_fwd | This study | PCR primer for KBP_L14 | TAGAGCCCCTAA CTGTAGCAGCA GGACCAGCAGC ATATCTGTTGGTCAAC |
| Sequence-based reagent | KBP_L14_rev | This study | PCR primer for KBP_L14 | GTTGACCAACAGAT ATGCTGCTGGTC CTGCTGCTACA GTTAGGGGCTCTA |
| Sequence-based reagent | KBP_L16_fwd | This study | PCR primer for KBP_L16 | TCCCTGAGAGACC CAGCAGCAGGAG CACCAGCAGGAGC AGGAGCAGGAGC AGCACGCCCTGCCATGTTA |
| Sequence-based reagent | KBP_L16_rev | This study | PCR primer for KBP_L16 | TAACATGGCAGGGC GTGCTGCTCCTGCT CCTGCTCCTGCTGGT GCTCCTGCTGCT GGGTCTCTCAGGGA |
| Sequence-based reagent | KBP_L18_fwd | This study | PCR primer for KBP_L18 | ATTGTTGATTACTG TGCAGCAGGACCA GGAGCCGCCCAGGAAATA |
| Sequence-based reagent | KBP_L18_rev | This study | PCR primer for KBP_L18 | TATTTCCTGGGCG GCTCCTGGTCCTGC TGCACAGTAATCAACAAT |
| Sequence-based reagent | KBP_HP4a_fwd | This study | PCR primer for KBP_L HP4a | ACTCATAACCTATA TGCACTAGCTGCA GTCTACCAGCATCTG |
| Sequence-based reagent | KBP_L HP4a _rev | This study | PCR primer for KBP_ HP4a | CAGATGCTGGTAG ACTGCAGCTAGT GCATATAGGTTATGAGT |
| Sequence-based reagent | KBP_HP4b_fwd | This study | PCR primer for KBP_L HP4b | AGTACACTAAAAC GCGCACTTGAGC ACAATGCC |
| Sequence-based reagent | KBP_L HP4b _rev | This study | PCR primer for KBP_ HP4b | GGCATTGTGCTCA AGTGCGCGTTT TAGTGTACT |
| Sequence-based reagent | KBP_HP5a_fwd | This study | PCR primer for KBP_L HP5a | GCTATCAATGCTGC TGCGTTGTCAGC GTTTTACATCAATAAG |
| Sequence-based reagent | KBP_ HP5a _rev | This study | PCR primer for KBP_ HP5a | CTTATTGATGTAAA ACGCTGACAACGC AGCAGCATTGATAGC |
| Sequence-based reagent | KIF15_FRB_fwd | This study | PCR primer for KIF15–FRB | AAGCTTGCCACCAT GGGCGCGCCTGCCAC CATGGCTCCTGG CTGCAAATCT |

*Continued on next page*

*Continued*

| Reagent type (species) or resource | Designation | Source or reference | Identifiers | Additional information |
|---|---|---|---|---|
| Sequence-based reagent | KIF15_FRB_rev | This study | PCR primer for KIF15–FRB | AGAGGATTCTAGAAG CAGGCGCGCCAGCG TAGTCTGGGACGTCG TATGGGTAGAATTCTC CTGGTGTCAGCTGCCCAGA |
| Sequence-based reagent | bioGFPKIF15 _fwd | This study | PCR primer for bioGFPKIF15 | AGCTCAAGCTTCGAA TTGGGCGCGCCAGCC ACCATGGCTCCTGG CTGCAAATCT |
| Sequence-based reagent | bioGFPKIF15_rev | This study | PCR primer for bioGFPKIF15 | GAATTCGATATCCTG CAGGTCGACTCCAG ATCCTCATCCTGGT GTCAGCTGCCCAGA |
| Sequence-based reagent | bioGFPKIF1A _fwd | This study | PCR primer for bioGFPKIF1A | TATTATAATGGCGCG CCAGCCACCGCCG GGGCCTCTGTGAAGGT |
| Sequence-based reagent | bioGFPKIF1A_rev | This study | PCR primer for bioGFPKIF1A | CTCGTCGACTCCTC CTCCTCATTTGGG AGAAAACACACCCAA |
| Commercial assay or kit | EnzChek Phosphate Assay Kit | Invitrogen | E6646 | |
| Chemical compound, drug | AP21967 | TaKaRa | Cat# 635057 | 1 µM |
| Chemical compound, drug | PEI | PolySciences | Cat# 24765–2 | |
| Chemical compound, drug | Fugene | Promega | Cat# E2692 | |
| Software, algorithm | ImageJ | NIH | https://imagej. nih.gov/ij/; RRID:SCR_003070 | |
| Software, algorithm | RELION | *Zivanov et al., 2018* | n/a | |
| Software, algorithm | CryoSparc2 | *Punjani et al., 2017* | n/a | |
| Software, algorithm | CisTEM | *Grant et al., 2018* | n/a | |
| Software, algorithm | MiRP | *Cook et al., 2020* | n/a | Protocol implemented in RELION |

## Protein expression and purification for cryo-EM

Full length human KBP residues 1–621 in a PSTCm1 expression vector (with kanamycin resistance and a N-terminal thrombin cleavable 6 × His-tag) was expressed in Rosetta2 cells (Novagen) as previously described (*Kevenaar et al., 2016*). Following immobilised metal-affinity chromatography with Ni-NTA resin (Qiagen), the 6 × His-tag was removed via incubation with thrombin protease overnight at 4°C. The protein was then subjected to reverse IMAC and further purified using size exclusion chromatography (SEC) into a buffer of 20 mM Tris-HCl (pH 7.4), 150 mM NaCl, 2.5 mM CaCl₂, and 1 mM DTT. Protein was snap-frozen and stored in at −80°C.

A human KIF15 motor domain and neck linker construct (residues 1–375) in a pET21a vector with a C-terminal 6 × His-tag was generated by chemical synthesis (GenScript, Piscataway, NJ). Six of the eight cysteine residues (C5S, C50S, C162S, C294S, C314S, and C346S) were mutated and two

cysteines were inserted (S250C and G375C) for orthogonal experiments not described further here. We refer to this construct as KIF15_MD6S. KIF15_MD6S was expressed and purified using methods previously described (Rosenfeld et al., 2005), then buffer exchanged into 25 mM HEPES pH 7.5, 100 mM KCl, 2 mM MgCl$_2$, 1 mM EGTA, 1 mM DTT, 0.1 mM ATP, snap frozen in liquid nitrogen, and stored at −80°C.

A human KIF1A motor domain and neck linker construct (KIF1A_MD residues 1–362) in a pFN18a vector (with a TEV protease-cleavable N-terminal Halo-tag and a C-terminal 6 × His-tag) was expressed in BL21-Gold (DE3) cells, as previously described (Atherton et al., 2014). Following a first IMAC with Ni-NTA resin, the Halo-tag was removed via incubation overnight with TEV protease at 4° C. The protein was then isolated from TEV via a second IMAC with Ni-NTA resin and further purified by SEC into a storage buffer of (20 mM HEPES, pH 7, 150 mM NaCl, 5 mM MgCl$_2$, 0.1 mM ADP, and 1 mM TCEP).

KIF15_MD6S or KIF1A_MD complexes with KBP were purified via IMAC using the 6 × His-tag on the kinesin constructs. Briefly, His-tagged kinesins were incubated with a 10 times excess of KBP in 20 mM Tris-HCl (pH 7.5), 150 mM NaCl, 1 mM MgCl$_2$, 10 mM Imidazole, 1 mM DTT, and 0.2 mM ADP for 5 min at 4°C. Following IMAC, complexes were eluted from the Ni-NTA resin (Qiagen) by addition of 200 mM imidazole, then dialysed at 4°C for 4 hr into 20 mM Tris-HCl (pH 7.5), 150 mM NaCl, 1 mM MgCl$_2$, 1 mM DTT, and 0.2 mM ADP.

## Steady-state ATPase assay

ATPase activity of KIF15_MD6S was measured in ATPase buffer (50 mM potassium acetate, 25 mM HEPES, 5 mM magnesium acetate, and 1 mM EGTA, pH 7.50) by measuring phosphate production in the presence of a minimum of a fivefold molar excess of paclitaxel-stabilised MTs, using a commercially available kit (EnzChek, Molecular Probes) at 20°C.

## Sample preparation for cryo-EM

KBP was prepared for cryo-EM using three different approaches. In the first approach, KBP was diluted to 0.15 mg/ml in KBP dilution buffer (20 mM Tris-HCl, pH 7.5, 150 mM NaCl, and 2 mM DTT) and 4 µl were applied to glow-discharged C-flat 2/2 holey carbon EM grids (Protochips, Morrisville, NC). For the second approach, KBP was diluted to 0.3 mg/ml in KBP dilution buffer and 4 µl were applied to glow-discharged 1.2/1.3 AuFoil gold grids (Quantifoil). For the third approach, glow-discharged C-flat 2/2 holey carbon EM grids were coated with graphene-oxide (GO) according to the protocol described by Cheng and colleagues (Cheng et al., 2020) and then 4 µl of KBP diluted to 0.02 mg/ml in KBP dilution buffer were added.

Kinesin motor domain–KBP complexes were diluted to 0.03 mg/ml in KBP–kinesin dilution buffer (20 mM Tris-HCl [pH7.5], 50 mM NaCl, 1 mM MgCl$_2$, 1 mM DTT, and 0.2 mM ADP) and 4 µl were added to the GO-coated gold grids described above. After a 30 s incubation of samples on the EM grid in a Vitrobot Mark IV (FEI Co., Hillsboro, OR) set at 4°C and 80% humidity, samples were blotted (6–8 s, blot force −10) and vitrified in liquid ethane. All steps were performed at 4°C.

For preparation of the KIF15_MD6S-MT complex, porcine tubulin (>99% pure, Cytoskeleton Inc) was polymerised in MES polymerisation buffer (100 mM MES, 1 mM MgCl$_2$, 1 mM EGTA, and 1 mM DTT, pH 6.5) with 5 mM GTP at 37°C and then stabilised with 1 mM paclitaxel. Approximately 70 µM KIF15_MD6S was pre-incubated for 5 min with 5 mM of AMPPNP in BRB80 at room temperature, and then mixed with 20 µM stabilised MTs. After a further incubation of 15 min, a 4 µl droplet was applied to a pre-glow discharged holey carbon grid (2/2 C-flat, Protochips Inc), blotted for 3.5 s, and then vitrified in liquid ethane using a Vitrobot Mark IV at ambient temperature and 80% humidity.

## Cryo-EM data collection

For dataset of KBP alone or KBP–KIF15_MD6S, low-dose movies were collected automatically using EPU software (Thermo Fisher, MA, USA) on a Titan Krios electron microscope (Thermo Fisher) operating at 300 kV, with a K2 summit direct electron detector (Gatan, CA, USA) and a quantum post-column energy filter (Gatan) operated in zero-loss imaging mode.

Datasets of KBP alone were collected either at eBIC or the ISMB, Birkbeck using a VPP, a sampling of ~1.05 Å/pixel and a nominal defocus range of 0.5–0.7 μm. The total dose was 42 e-/Å$^2$ over 40 frames, with the detector operating in counting mode at a rate of ~5 e-/pixel/s.

Datasets of KBP–kinesin complexes were collected at the ISMB, Birkbeck without a phase plate and a nominal defocus range of 1.5–4 μm. KIF1A_MD–KBP complexes were collected at a sampling of 0.85 Å/pixel, whereas KBP–KIF15_MD6S complexes were collected at a sampling of 1.047 Å/pixel. For KIF1A_MD–KBP complexes, the total dose was 88 e-/Å$^2$ over 36 frames, with the detector operating in counting mode at a rate of 7.1 e-/pixel/s. For KBP–KIF15_MD6S complexes, the total dose was 80 e-/Å$^2$ over 64 frames, with the detector operating in counting mode at a rate of 5.7 e-/pixel/s.

The KIF15_MD6S-MT dataset was collected manually on a Tecnai Polara microscope (Thermo Fisher) at the ISMB, Birkbeck, operating at 300 kV, with a K2 summit direct electron detector (Gatan, CA, USA) and a quantum post-column energy filter (Gatan) operated in zero-loss imaging mode. A nominal defocus range of 1.0–3.5 μm and a final pixel size of 1.39 Å was used. The total dose was 32 e-/Å$^2$ over 50 frames, with the detector operating in counting mode at a rate of 6.2 e-/pixel/s.

## Cryo-EM data processing

Low-dose movies were motion-corrected using MotionCor2 (*Zheng et al., 2017*) with a patch size of 5, generating full-dose and dose-weighted sums. CTF determination was performed on full-dose sums with gCTF (*Zhang, 2016*) and then dose-weighted sums were used for all further processing. Data were cleaned at this stage by first excluding all micrographs with gCTF resolutions worse than 4.5 Å, as estimated with a custom cross-correlation coefficient cutoff (Python script kindly shared by Radostin Danev), then manually removing micrographs with poor appearance (ice contamination, protein aggregation, or poor GO coverage) in real or reciprocal space. For KBP alone data, micrographs with calculated phase shifts outside the expected phase shift progression at each plate position were also excluded.

Particles were first picked using Eman2's neural network picker (*Bell et al., 2018*), with a 180 pixel box size for KBP-alone and KBP–KIF15_MD6S datasets, or a 220 pixel box size for the KBP–KIF1A_MD datasets. Good 2D classes were then used as templates to pick the data with Gautomatch (http://www.mrc-lmb.cam.ac.uk/kzhang/).

For Eman2 neural network picker or Gautomatch-derived particles from each dataset, separate multiple rounds of 2D classification were performed in RELION v3.0 (*Zivanov et al., 2018*), cryoSPARC2 (*Punjani et al., 2017*), or cisTEM (*Grant et al., 2018*). This resulted in a total of six sets of good 2D classes showing clear secondary structure for each dataset, two produced by each programme for each picking method. For each dataset, these six good 2D class sets for each dataset were then combined and duplicate particles removed. At this stage, for each sample (KBP-alone, KBP–KIF15_MD6S, or KBP–KIF1A_MD) good 2D classes from their constituent datasets were combined.

KBP–KIF15_MD6S or KBP–KIF1A_MD datasets composed of their respective constituent datasets were easily combined, being from the same microscope and optical set up. However, KBP-alone data were collected on different microscopes and had a range of pixel sizes (<2% difference). KBP-alone data therefore was combined at this stage using the optics grouping protocol in RELION v3.1 (*Zivanov et al., 2018*).

KBP-alone and KBP–KIF15_MD6S data were taken to 3D processing at this stage, while multiple attempts to process KBP–KIF1A_MD data in 3D gave no reliable results. For KBP-alone and KBP–KIF15_MD6S data, de novo initial 3D models were created in cryoSPARC2. For KBP–KIF15_MD6S data, a single round of 3D classification was performed in RELION v3.0 and the best class selected and auto-refined. For KBP-alone data, 3D classification in RELION v3.1 or cryoSPARC2 did not reveal different 3D structures or improve reconstructions over sorting only in 2D; therefore, particles selected with 2D classification were used as direct input for auto-refinement. The final KBP-alone map was sharpened with a B-factor of −200 to the gold-standard Fourier shell correlation (FSC) 0.143 cutoff (4.6 Å). The KBP–KIF15_MD6S map was sharpened locally with a B-factor of −495, according to local resolutions determined using RELION v3.1's inbuilt local resolution software.

The KIF15_MD6S-MT dataset was processed using our MT RELION-based pipeline (MiRP) as described previously, using low-pass filtered KIF5B_MD-decorated MTs as references (*Atherton et al., 2019*; *Cook et al., 2020*). KIF15_MD6S 13-protofilament-MTs were the most

common MT architecture and were selected after supervised 3D classification in MiRP for analysis. The symmetrised asymmetric unit (KIF15_MD6S plus a tubulin dimer) was locally sharpened in UCSF Chimera with a B-factor of −134 according to local resolutions determined using RELION v3.0's inbuilt local resolution software.

All displayed 3D molecular representations were made in UCSF Chimera or ChimeraX software (*Pettersen et al., 2004*; *Goddard et al., 2018*). Data collection and model refinement statistics can be found in *Table 1*.

## Cryo-EM model building and refinement

Due to low overall homology to available structures in the protein data bank (PDB), structure prediction of KBP produced poor models with little resemblance to the cryo-EM density. KBP was therefore modelled using a combination of secondary structure prediction, TPR prediction, fragment homology information, prior knowledge of right-handed alpha-solenoid proteins, and with reference to the cryo-EM density.

TPR motifs were identified in the KBP sequence using the TPRpred server (*Karpenahalli et al., 2007*) available in the MPI Bioinformatics Toolkit (*Zimmermann et al., 2018*). Secondary structure predictions using Raptor X (*Wang et al., 2016b*), iTasser (*Roy et al., 2010*), JPred (*Drozdetskiy et al., 2015*), Spider2 (*Yang et al., 2017*), PSSpred (*Yan et al., 2013*), and SOPMA (*Geourjon and Deléage, 1995*) were then run on the sequence and consensus between these multiple predictions used to assign likely α-helical content. To identify regions dispensable for the overall fold and likely disordered loop regions, disorder prediction was performed with Raptor X and inter-species low homology regions in KBP (from early Metazoans to humans) were determined via Clustal Omega multiple sequence alignment (*Sievers et al., 2011*). Finally, weak homology models for overlapping fragments of the structure were identified using the HHpred (*Hildebrand et al., 2009*) server in the MPI Bioinformatics Toolkit.

With the information described above a sequence alignment was built with KBP and the following fragment homology model PDBs; 5OJ8, 4A1S, 3QC1, 4NQ0, 4AIF, and 5M × 5. This sequence alignment was used as a basis for multiple rounds of modelling and flexible fitting with Modeller (*Sali and Blundell, 1993*) and Flex-EM (*Topf et al., 2008*) respectively, using α-helical secondary structure restraints. This modelling process was guided by consistency with the cryo-EM density and secondary structure and TPR predictions described above. Finally, the structure was refined against the cryo-EM density in real-space with five macro-cycles in Phenix (*Afonine et al., 2018*). All 19 predicted and modelled helices were accounted for by rod-like cryo-EM density in the reconstruction and at 4.6 Å resolution, density was discernible for bulky side chains in the TPR regions (*Figure 1—figure supplement 1a and b*), providing a validation of the assigned sequence directionality in the fold.

The KBP–KIF15_MD6S model was built as follows: the final KBP model described above and the KIF15_MD X-ray crystallographic model (PDB code:4BN2 *Klejnot et al., 2014*) were rigid fitted into the KBP–KIF15_MD6S density in Chimera. Density for an extended α6-helix and docked neck-linker in KIF15_MD6S were absent; therefore, Modeller was used to model a short α6-helix and the neck-linker removed. A model for the L11, α4-helix, and L12 region in KIF15_MD6S were then created using Coot (*Emsley and Cowtan, 2004*) and Modeller. The model was refined into the cryo-EM density in real-space using Phenix (*Afonine et al., 2018*) with secondary structure restraints. A first refinement of 15 cycles used rigid bodies describing α-helical hairpins in KBP to get a good rough fit. Following this, the whole complex was further refined without rigid bodies for another five macro-cycles.

The KIF15_MD6S–MT model was built as follows: the KIF15_MD, KIF11_MD, and KIF5B_MD-tubulin X-ray crystallographic models (PDB codes:4BN2 *Klejnot et al., 2014*; *Gigant et al., 2013*; *Parke et al., 2010*) were used as homology models in Modeller to build the KIF15 part of the complex. The KIF15_MD6S model and the paclitaxel–MT tubulin dimer model (*Kellogg et al., 2017*) were then rigid fitted into KIF15_MD6S–MT density, combined then refined in real-space with five macro-cycles in Phenix with peptide backbone restraints.

## Antibodies, reagents, and expression constructs for cell biology

The following antibodies were used for immunofluorescence staining: mouse anti-HA (1:500, Roche) and goat-anti-mouse Alexa 488 (1:400, Thermo Fisher Scientific). The following antibodies were used for western blot: mouse anti-HA (1:2,000, BioLegend), rabbit anti-GFP (1:10,000, Abcam), goat anti-mouse IRDye800CW (1:15.000, LI-COR), and goat-anti-rabbit IRDye680LT (1:20,000, LI-COR). A reagent used in this study is rapalog (AP21967, TaKaRa).

The following DNA expression constructs in this study have been described before: GW1–PEX3–mRFP–FKBP1, β-actin–Kif1A_MDC–FRB (*Kevenaar et al., 2016*) (mouse cDNA), BirA coding vector (*van der Vaart et al., 2013*), and pebioGFP (*van der Vaart et al., 2013*). pGW1–HA–KBP contained a linker (GGATCCCCGGAATTCGGCACGAGGGAGGCCGCT) between the HA tag and KBP and was cloned using PCR-based strategies with human KBP cDNA (KIAA1279, IMAGE clone 4550085) as template and ligation into the pGW1–HA backbone. A similar strategy was used to generate the mutated KBP constructs, listed in *Table 2*. β-actin–KIF15_MDC–FRB was cloned using a PCR-based Gibson Assembly strategy with mouse KIF15 cDNA as template into the β-actin–KIF1A_MDC–FRB backbone. PebioGFP-KIF1A_MDC and pebioGFP-KIF15_MDC were cloned into the pebioGFP backbone using PCR-based strategies with MDC–FRB constructs as templates.

## Cell culture, transfection, and immunofluorescence staining

COS-7 cells were purchased from ATCC and routinely checked for mycoplasma contamination using LT07-518 Mycoalert assay (Lonza). Cells were cultured in 50/50 DMEM (Lonza)/Ham's F10 (Lonza) medium supplemented with 10% FCS (Sigma) and 1% penicillin/streptomycin (Sigma). One day before transfection cells were diluted and plated on 18 mm glass coverslips. COS-7 cells were transfected using FuGENE6 (Roche) following the manufacturer's protocol. Next day, rapalog (final concentration 1 μM) was added and cells were incubated for 3 hr. Cells were then fixed with 4% formaldehyde/4% sucrose in phosphate-buffered saline (PBS) for 10 min at room temperature, washed three times PBS-CM (PBS supplemented with 1 mM $MgCl_2$ and 0.1 mM $CaCl_2$), permeabilised in 0.2% TritonX-100 for 15 min, and washed one time with PBS-CM. Cells were first incubated with 0.2% gelatin for 30 min at 37°C, and then with primary antibodies, diluted in 0.2% gelatin, for 30 min at 37°C. After washing three times with PBS-CM, cells were incubated for 30 min at 37°C with secondary antibody diluted in 0.2% gelatin, washed three times in PBS-CM, and finally mounted using Fluoromount (Invitrogen).

**Table 2.** Kinesin-binding protein (KBP) mutants used in this study.

The original and mutated amino acid (top) and nucleotide sequences (bottom) are shown for each construct.

| Construct | Original sequence | Mutated to |
|---|---|---|
| L1 | EKEPYK gagaaggaaccatacaag | AAAPAA gcagcagcaccagcagca |
| L3 | TEE acggaggag | AGG gcaggagga |
| L5 | REE agagaagaa | AGA gcaggagca |
| L10 | KISATEDTPEAEGEVPEL aagatctcagccacagaagacactc ctgaagctgaaggagaagtgccagagctt | AGAGAGAGPAGAGAGPGG gcaggagcaggagcaggagca ggaccagcaggagcaggagcaggaccaggagga |
| L12 | DGY gatggttat | GGA ggcggcgcg |
| L14 | DLNPQY gacctgaatccacagtat | AAGPAA gcagcaggaccagcagca |
| L16 | NKVFPEHIGEDVL aataaagtattccctgagcata tagggggaagatgttctt | AAGAPAGAGAGAA gcagcaggagcaccagcag gagcaggagcaggagcagca |
| L18 | EKHPE gaaaagcatcctgag | AAGPG gcagcaggaccagga |
| αHP4a | YLAQ tacctagctcaa | ALAA gcactagctgca |
| αHP4b | Q cag | A gca |
| αHP5a | TLSQ accttgtcacag | ALSA gcgttgtcagcg |

## Cell biology image analysis and quantification

Fixed cells were imaged on a Carl Zeiss LSM 700 confocal laser scanning microscope running ZEN2011 software, using a Plan-Apochromat 40×/1.30 oil DIC objective and image settings were maintained the same for all images within one experiment. Images were acquired of cells that express similar levels of HA–KBP constructs based on immunostaining (*Figure 5—figure supplement 1*). Cells were selected on a first come first served basis. Images were processed and analysed using Fiji software (*Schindelin et al., 2012*). To calculate the percentage of cells in which translocation of peroxisomes was observed, imaged cells were classified as either translocating, when peroxisomes re-localised into the cell periphery, or not translocating, when peroxisomes remained in the cell centre. For quantification of PEX translocation, an ROI of the cell area was drawn and from this a second ROI at 5 (KIF1A_MDC) or 7.5 (KIF15_MDC) µm from the outer cell area was created. Images were thresholded at 7500 (KIF15_MDC) or 10,000 (KIF1A_MDC). Different peripheral areas and threshold values were defined for the two kinesins, due to observed differences in translocation properties between the kinesins (compare *Figure 5—figure supplement 3a–c*). For the two selected ROIs, the area with fluorescent intensity above threshold was determined in the RFP channel. From these values the percentage of cell area above threshold in the cell periphery from the total area above threshold was calculated.

## Pull-down experiments and western blotting

HEK293T cells were purchased from ATCC and routinely checked for mycoplasma contamination using LT07-518 Mycoalert assay (Lonza). Cells were cultured in 50/50 DMEM (Lonza)/Ham's F10 (Lonza) medium supplemented with 10% FCS (Sigma) and 1% penicillin/streptomycin (Sigma). One day before transfection cells were diluted and plated into 6-well plates. Cells were co-transfected with pCl-Neo-BirA, HA-tagged constructs, and bioGFP-tagged constructs using MaxPEI (Polysciences) in a ratio of 3/1 PEI/DNA, according to the manufacturer's protocol. After 24 hr of expression, cells were washed in ice-cold PBS and lysed in lysis buffer (100 mM Tris-HCl pH 7.5, 150 mM NaCl, 1% Triton X-100, and protease inhibitors [Roche]) for 30 min on ice. Lysates were cleared by 30 min centrifugation at 13.2 krpm at 4˚C and supernatants were incubated with blocked (incubation for 30 min at RT in 50 mM Tris-HCl pH 7.5, 150 mM KCl, and 0.2 µg/µl chicken egg albumin) Streptavidin Dynabeads M-280 (Invitrogen) for 1.5 hr at 4˚C. Beads were then washed five times with washing buffer (100 mM Tris-HCl pH 7.5, 250 mM NaCl, and 0.5% Triton X-100) and proteins were eluded from the beads by boiling for 10 min at 95˚C in 2× DTT+sample buffer (20% glycerol, 4% SDS, 200 mM DTT, 100 mM Tris-HCl pH 6.8, and bromophenol blue).

Protein samples were run on 10% SDS-PAGE gels and transferred to nitrocellulose membranes (Bio-Rad) by semi-dry blotting at 16V for 1 hr. Membranes were blocked by incubation in 3% bovine serum albumin (BSA) in PBST (PBS supplemented with 0.02% Tween20) for 1 hr at room temperature. This was followed by overnight incubation with primary antibodies in 3% BSA-PBST. Membranes were washed three times with PBST, incubated with secondary antibody in 3% BSA-PBST for 1 hr at room temperature, and washed three times with PBST. Membranes were scanned using an Odyssey Infrared Imaging system (LI-COR Biosciences) and blots were acquired at 680 nm and 800 nm.

## Acknowledgements

JA was supported by a grant from the Medical Research Council (MRC), UK (MR/R000352/1) to CAM, and JL and AP were supported by a grant from Worldwide Cancer Research, UK (16–0037) awarded to JL and CAM. We thank Dr Alexander Cook for technical and processing assistance at the ISMB and Dr Radostin Danev at the Graduate School of Medicine, The University of Tokyo for custom processing scripts. Cryo-EM data collected at the Institute of Structural and Molecular Biology (ISMB), Birkbeck was on equipment funded by the Wellcome Trust, UK (202679/Z/16/Z, 206166/Z/17/Z and 079605/Z/06/Z) and the Biotechnology and Biological Sciences Research Council (BBSRC), UK (BB/L014211/1). We thank Dr Natasha Lukoyanova for support during data collection at the ISMB. For the remaining EM data collection, we acknowledge Diamond for access and support to the Electron Bioimaging Centre (eBIC) at Diamond, Harwell, UK, funded by the MRC, BBSRC and Wellcome Trust, UK. SSR was supported by a grant from the National Institute of General Medical

Sciences (R01GM130556). NO and MOS were supported by a grant awarded to MOS from the Swiss National Science Foundation (31003A_166608). JJA was supported by the Netherlands Organization for Scientific Research (NWO-ALW-VICI, CCH) and the European Research Council (ERC) (ERC-consolidator, CCH).

## Additional information

### Funding

| Funder | Grant reference number | Author |
|---|---|---|
| Medical Research Council | MR/R000352/1 | Joseph Atherton<br>Carolyn A Moores |
| Worldwide Cancer Research | 16-0037 | Julia Locke<br>Alejandro Peña<br>Carolyn A Moores |
| Wellcome Trust | 202679/Z/16/Z | Carolyn A Moores |
| Biotechnology and Biological Sciences Research Council | BB/L014211/1 | Carolyn A Moores |
| National Institute of General Medical Sciences | R01GM130556 | Steven S Rosenfeld |
| Swiss National Science Foundation | 31003A_166608 | Natacha Olieric<br>Michel O Steinmetz |
| Netherlands Organization for Scientific Research | NWO-ALW-VICI | Jessica JA Hummel<br>Casper C Hoogenraad |
| European Research Council | ERC-consolidator | Jessica JA Hummel<br>Casper C Hoogenraad |
| Wellcome Trust | 206166/Z/17/Z | Carolyn A Moores |
| Wellcome Trust | 079605/Z/06/Z | Carolyn A Moores |
| Netherlands Organization for Scientific Research | CCH | Jessica JA Hummel<br>Casper C Hoogenraad |
| European Research Council | CCH | Jessica JA Hummel<br>Casper C Hoogenraad |

The funders had no role in study design, data collection and interpretation, or the decision to submit the work for publication.

### Author contributions

Joseph Atherton, Conceptualization, Resources, Data curation, Formal analysis, Validation, Investigation, Visualization, Methodology, Writing - original draft, Project administration, Writing - review and editing, Corresponding author; Jessica JA Hummel, Conceptualization, Resources, Data curation, Formal analysis, Validation, Investigation, Visualization, Methodology, Writing - original draft, Writing - review and editing; Natacha Olieric, Conceptualization, Resources, Writing - review and editing; Julia Locke, Alejandro Peña, Resources, Investigation, Writing - review and editing; Steven S Rosenfeld, Michel O Steinmetz, Resources, Funding acquisition, Writing - review and editing; Casper C Hoogenraad, Conceptualization, Resources, Supervision, Funding acquisition, Project administration, Writing - review and editing; Carolyn A Moores, Conceptualization, Resources, Supervision, Funding acquisition, Visualization, Writing - original draft, Project administration, Writing - review and editing

### Author ORCIDs

Joseph Atherton https://orcid.org/0000-0002-6362-2347
Casper C Hoogenraad http://orcid.org/0000-0002-2666-0758
Carolyn A Moores http://orcid.org/0000-0001-5686-6290

### Decision letter and Author response

Decision letter https://doi.org/10.7554/eLife.61481.sa1

Author response https://doi.org/10.7554/eLife.61481.sa2

## Additional files

### Supplementary files

• Transparent reporting form

### Data availability

Cryo-EM electron density maps and models have been deposited in the electron microscopy data bank (EMDB) and the protein data bank (PDB), respectively. The relevant deposition codes are provided in Table 1.

The following datasets were generated:

| Author(s) | Year | Dataset title | Dataset URL | Database and Identifier |
|---|---|---|---|---|
| Atherton J, Hummel JJA, Olieric N, Locke J, Peña A, Rosenfeld SS, Steinmetz MO, Hoogenraad CC, Moores CA | 2020 | Kinesin binding protein (KBP) | http://www.rcsb.org/structure/6ZPG | RCSB Protein Data Bank, 6ZPG |
| Atherton J, Hummel JJA, Olieric N, Locke J, Peña A, Rosenfeld SS, Steinmetz MO, Hoogenraad CC, Moores CA | 2020 | Kinesin binding protein complexed with Kif15 motor domain | http://www.rcsb.org/structure/6ZPH | RCSB Protein Data Bank, 6ZPH |
| Atherton J, Hummel JJA, Olieric N, Locke J, Peña A, Rosenfeld SS, Steinmetz MO, Hoogenraad CC, Moores CA | 2020 | Kinesin binding protein complexed with Kif15 motor domain. Symmetrised asymmetric unit. | http://www.rcsb.org/structure/6ZPI | RCSB Protein Data Bank, 6ZPI |
| Atherton J, Hummel JJA, Olieric N, Locke J, Peña A, Rosenfeld SS, Steinmetz MO, Hoogenraad CC, Moores CA | 2020 | Kinesin binding protein (KBP) | https://www.ebi.ac.uk/pdbe/emdb/EMD-11338 | Electron Microscopy Data Bank, EMD-11338 |
| Atherton J, Hummel JJA, Olieric N, Locke J, Peña A, Rosenfeld SS, Steinmetz MO, Hoogenraad CC, Moores CA | 2020 | Kinesin binding protein complexed with Kif15 motor domain | https://www.ebi.ac.uk/pdbe/emdb/EMD-11339 | Electron Microscopy Data Bank, EMD-11339 |
| Atherton J, Hummel JJA, Olieric N, Locke J, Peña A, Rosenfeld SS, Steinmetz MO, Hoogenraad CC, Moores CA | 2020 | Kinesin binding protein complexed with Kif15 motor domain. Symmetrised asymmetric unit. | https://www.ebi.ac.uk/pdbe/emdb/EMD-11340 | Electron Microscopy Data Bank, EMD-11340 |

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
