## [Decision Letter]

**Acceptance summary:**

Kinesin-binding protein (KBP) is an inhibitory regulator of certain kinesin family members. In this manuscript, you use cryo-EM to solve the structure of KBP on its own and in complex with the Kif15 kinesin motor domain. KBP binds and remodels the tubulin-binding interface of the kinesin, preventing interaction with the microtubule. Mutagenesis experiments validate the mechanism described by the structure, and conservation between KBP-sensitive kinesins is used to describe specificity of KBP binding.

**Decision letter after peer review:**

Thank you for submitting your article "The mechanism of selective kinesin inhibition by kinesin binding protein" for consideration by *eLife*. Your article has been reviewed by Cynthia Wolberger as the Senior Editor, a Reviewing Editor, and three reviewers. The following individuals involved in review of your submission have agreed to reveal their identity: Hauke Drechsler (Reviewer #2); Hernando Sosa (Reviewer #3).

The reviewers have discussed the reviews with one another and the Reviewing Editor has drafted this decision to help you prepare a revised submission.

Summary:

Kinesin-binding protein (KBP) is an inhibitory regulator of certain kinesin family members. In this manuscript, Atherton et al. use cryo-EM to solve the structure of KBP on its own and in complex with the Kif15 kinesin motor domain. KBP alone forms a TPR-containing α-solenoid, which unfurls to wrap around the Kif15 motor domain in the complex structure. By binding to (and remodeling) the tubulin-binding interface of the kinesin, KBP prevents any interaction between the kinesin and the microtubule, thus explaining the mechanism of inhibition. Mutagenesis experiments validate the mechanism described by the structure, and conservation between KBP-sensitive kinesins is used to describe specificity of KBP binding.

Essential revisions:

1) The claim made by the title concerning kinesin selectivity is a bit too bold. Resolving the structure allows the authors to put forward a hypothesis about the mechanism of KBP-selectivity, but this still needs to be tested/validated. Please adjust the title.

2) It is a bit odd that the authors used Cys-substituted rather than wild-type KIF15 for their study and that they do mention this fact in the Material and methods section only. The authors should disclose this properly in the text (maybe the authors could name their construct something like KIF156S (?) or similar throughout the text, to avoid any misconception) and discuss possible caveats associated with these substitutions. In particular:

A) Although, many studies of kinesin proteins have used cys-substituted constructs, at least one study (Andreasson et al., 2015) has reported functional differences between a kinesin wild type and the cys-substituted version.

B) Could the substitutions affect KBP binding? One substitution is in kinesin helix 4 which is in the middle of the KBP binding site.

C) Could the mutations facilitate the large conformational change observed in KIF15? e.g. by allowing unraveling of the loops connecting the kinesin H4 helix to the rest of the motor domain?

D) The KIF15 motor domain construct is cys-substituted but the KIF1A construct is not. Could the differences observed between the KIF1A-KBP and KIF15-KBP complexes be partially related to the mutations introduced in the KIF15 motor domain?

3) The experimental setup of data shown in Figure 5 and Figure 5—figure supplement 1 is inconsistent and not well explained (see A-D below). Experiments are not sufficiently controlled (E-H below) and data of main figure and figure supplement is at least partially contradictive (I-J below). The authors need to re-analyse existing data and provide some new data. More detail in the sub-points below.

A) Figure 5 e.g: How is "transport" defined here? Is transport = some mRFP intensity over threshold within the 5 µm from cell perimeter? Otherwise it is hard to believe that a biological system, which always shows some background activity, produces absolute '0' values in this assay. I would kindly ask the authors to provide a proper definition of "transport" to the reader here.

B) Figure 5: I assume that the assay has been performed in interphase cells. KIF15 however does not localise to microtubules in interphase cells but rather to actin stress fibres (Buster et al., 2006), presumably by interaction with myosin 2B (Feng et al., 2016). Theoretically, some of the peroxisome behaviour observed here could therefore be actin/myosin dependent as well. This is probably not the case since the myosin 2B interaction was mapped to the KIF15-tail, which is missing here and the motor domain is probably overexpressed, uncoupling it form endogenous targeting/regulation mechanisms. However, to validate the assay in the first place, it should have been tested briefly, whether the KIF15 MD localises to microtubules at all.

C) Figure 5 D/F: As far as I understand it, the authors define an intensity threshold and quantify the cell area within 7.5 or 5 µm from the cell perimeter that is 'covered' by intensity larger a certain threshold. Why 5 µm in one case and 7.5 µm in the other case? Is thresholding really a good idea here? Given a fixed total-intensity-over-threshold equally distributed along the 5 µm (center to periphery) or with a distribution highly skewed towards the periphery. The skewed distribution would indicate better transport but would cover less area. Would something like fraction from overall intensity (above threshold) in a certain distance from the cell perimeter be more accurate?

D) Please provide an explanation (e.g. in the legend or methods) for why the quantification thresholds are different for the two kinesins in Figure 5D/F. For Kif15 the periphery is defined as the outer 7.5μm of the cell, but for Kif1A it is 5μm. Does this threshold value affect the interpretation of the results?

E) Figure 5: KBP inhibits KIF15 motor domains in a 1:1 stoichiometry. The authors kind of 'control' for the KBP expression (subsection “Cell biology image analysis and quantification”: 'Images were acquired of cells that express similar levels of HA-KBP constructs based on immunostaining'), but not for the KIF15/KIF1A-MD levels. KIF15/KIF1A-MD levels might change from mutant to mutant due to different stability in vivo (and from cell to cell, transient expression). Hence, we cannot formally exclude that a drop in peroxisome transport could as well be caused by [c] KBP > [c] KIF15 MD, masking positive (i.e., disruptive) hits or that false positive hits would be created when [c] KIF15 MD > [c] KBP.

Thus, the authors should please provide a quantification of the KIF15-MD expression, allowing the reader to estimate, whether the KIF15/KBP ratios are the same throughout the experiment. Since the authors selected single cells based on the KBP signal, KIF15-MD quantification should also be done per cell.

F) Figure 5—figure supplement 1: The authors compared the binding of WT and mutant KBP variants by quantifying the amount of HA-KBP that coprecipitated together with bioGFP-KIF15-MD. The amount is given as "intensity KBP-PD/intensity KBP input" normalised to the WT KBP condition. However, it is not clear whether the PD/Input quotient has been calculated first and then been normalised to WT KBP or if the quotient has been calculated from already normalized values. First case would not be ok, as Input and PD were clearly run and quantified on different gels and are therefore obviously not directly comparable – e.g., the pull-down bands for KBP to L10m in the first panels of (b) and (c) are stronger than their corresponding input bands.

However, I did a quick quantification (ImageJ) of the blots shown in the left panel of (b) normalising each blot to WT KBP first. Still more protein was pulled down than has been put into the assay in the first place – how can this be? Also. Doing it like this, L12m would not be a positive hit any longer. While trying to quantify the other blots, I recognized that the tonal ranges for the KIF15-MD blots (b, right panel), (c, both panels) and the KBP input blot in (c, left) has massively been pushed to the higher intensities, leaving virtually no background in the lanes above and below the bands. If this happened during figure preparation – please don't do that. If this already happened due to wrong settings during data acquisition in the imager, your quantifications might not have bene accurate.

I recommend to (re-)run both input and PD sample together on the same gel (there are 26-well gels commercially available). Like this they are treated the same way during western blotting/detection – allowing a more reliable cross comparison.

G) Figure 5—figure supplement 1: comparing the PD/Input quotient only makes sense, if we assume that KIF15/ KIF1A-MD pulldown occurs from all lysates with the same efficiency and the overall KIF15/KIF1A-MD amount is not limiting KBP pulldown. This, however, is clearly not the case: in (c) lanes L14m, L16m, L18m, L12m+L14m and alphaHP4am the amount of pulled down KIF1A-MD is much less than for the other constructs, while the KBP input in comparison is even higher. Less KIF15/KIF1A-MD in the pulldown fraction might reflect a weaker KIF15/KIF1A-MD expression (mutant stability in vivo, transient transfection), less efficient biotinylation by BirA (transiently transfected as well) in vivo or less efficient pulldown by the streptavidin beads – we just can't tell from these figures. Nevertheless, if just 10% of the total KIF15/KIF1A-MD are pulled down, only 10% of total KBP should be expected to co-precipitate as well.

Hence, the authors should please show and quantify the KIF15/KIF1A-MD input as well and relate the fraction of co-precipitated KBP with the fraction of pulled down motor domain (i.e., are mutants underperforming regarding their expected pulldown efficiency?).

I would also like to encourage the authors to verify their positive hits (i.e., L12m, L14m, HP4am, HP4bm and HP5m), by co-precipitating defined amounts of recombinant KBP/KIF-MD proteins to be on the safe side. Since expression was done in bacteria and purification protocols are established this could be done in a reasonable amount of time.

H) In Figure 5—figure supplement 1C, the intensity of the bait (bioGFP-Kif1A_MDC) appears to be significantly lower for the L16m and L18m lanes compared to the other lanes. A lower level of bait could explain the weaker pulldown of HA-KBP in these lanes. The data should be re-quantified to take into account the amount of bait pulled down, or the experiment should be repeated with a consistent level of bait across all lanes (as in Figure 5—figure supplement 1). Since the authors highlight the weaker interaction between the L18m KBP and Kif1A compared to Kif15 in the text, this is an important control to include.

I) Figure 5 and Figure 5—figure supplement 1: The L10m and L16m mutation bind KIF15-MD like WT KBP, but fail to inhibit peroxisome transport. If this effect is real (see points above), the authors should please comment on these results, since they are in conflict with their simple binding = inhibition model.

J) The text doesn't address inconsistencies between the results of the peroxisome assay and the immunoprecipitation. For example, in Figure 5D/E, the mutation to loops 10 and 16 causes the peroxisomes to disperse to the periphery, which indicates that the inhibition of kinesin by KBP is relaxed. However, in Figure 5—figure supplement 1B the interaction between KBP and the kinesin does not appear to be significantly lower (and in the case of L10 is stronger than the control). These results appear to be directly contradictory. Can the authors provide an explanation for this effect?

---

## [Author Response]

Essential revisions:1) The claim made by the title concerning kinesin selectivity is a bit too bold. Resolving the structure allows the authors to put forward a hypothesis about the mechanism of KBP-selectivity, but this still needs to be tested/validated. Please adjust the title.

We propose the alternative title: “The mechanism of kinesin inhibition by kinesin binding protein", and we have adjusted the text throughout the rest of the manuscript (e.g. in the Abstract) to reflect that we are proposing a hypothesis concerning kinesin selectivity.

2) It is a bit odd that the authors used Cys-substituted rather than wild-type KIF15 for their study and that they do mention this fact in the Material and methods section only. The authors should disclose this properly in the text (maybe the authors could name their construct something like KIF156S (?) or similar throughout the text, to avoid any misconception)

In addition to what was already included in the Material and methods section, we have now included this information in the Results section, and use the abbreviation KIF15_MD6S throughout the manuscript.

and discuss possible caveats associated with these substitutions. In particular:A) Although, many studies of kinesin proteins have used cys-substituted constructs, at least one study (Andreasson et al., 2015) has reported functional differences between a kinesin wild type and the cys-substituted version.

We now include new steady state ATPase activity of KIF15_MD6S in Figure 2—figure supplement 1A: its kcat = 2.9 ± 0.5 s-1 and K0.5,MT = 4.8 ± 1.4 uM, which are very similar to published reports of WT KIF15 (1-375): kcat = 2.1 s-1 and K0.5,MT = 3.1 μm (Klejnot et al., 2013; PMID 24419385). ATPase-related text has also been added to the Materials and methods section. While we agree that such protein engineering might alter motor function, this is not a general rule. The additional data included (and structural data already present in our manuscript, see more below) provide evidence that this is not the case for KIF15_MD6S.

B) Could the substitutions affect KBP binding? One substitution is in kinesin helix 4 which is in the middle of the KBP binding site.C) Could the mutations facilitate the large conformational change observed in KIF15? e.g. by allowing unraveling of the loops connecting the kinesin H4 helix to the rest of the motor domain?

We cannot completely exclude these possibilities but do not think they are likely because:

i) kinesin helix 4 is also part of the MT binding site – we show in our KIF15_MD6S-MT reconstruction (Figure 3—figure supplement 1F) that the density for the H4 helix is very clear and forms a canonical interaction with the MT.

ii) this KIF15_MD6S-MT reconstruction also shows that AMPPNP produces a canonical conformational change in the motor domain, supporting the idea that mechanochemical response of the KIF15_MD6S protein is not disrupted. We have included additional views of this reconstruction in Figure 3—figure supplement 3 to support this point. This is also supported by the ATPase data described in response 3.

iii) The alignments in Figure 6A show that the cysteine residues in KBP binding regions of Kif15 are not strongly conserved amongst KBP-binding kinesins.

We have now included these points in the Discussion.

D) The KIF15 motor domain construct is cys-substituted but the KIF1A construct is not. Could the differences observed between the KIF1A-KBP and KIF15-KBP complexes be partially related to the mutations introduced in the KIF15 motor domain?

Again, we cannot completely exclude this possibility. However, as we describe in the manuscript, there are a number of more obvious differences between these two samples relating to the behaviour of the complexes on the cryo-EM grids that are more likely to explain the observed differences. We have now also included this caveat in the Discussion.

3) The experimental setup of data shown in Figure 5 and Figure 5—figure supplement 1 is inconsistent and not well explained (see A-D below). Experiments are not sufficiently controlled (E-H below) and data of main figure and figure supplement is at least partially contradictive (I-J below). The authors need to re-analyse existing data and provide some new data.

We now include more data and data re-analysis as suggested by the reviewer, described in detail below. We have also completely restructured this section of the text to enable clearer cross-comparison of the mutant and motor behaviour in the context of each assay.

More detail in the sub-points below.A) Figure 5 e.g: How is "transport" defined here? Is transport = some mRFP intensity over threshold within the 5 µm from cell perimeter? Otherwise it is hard to believe that a biological system, which always shows some background activity, produces absolute '0' values in this assay. I would kindly ask the authors to provide a proper definition of "transport" to the reader here.

This is a good point. Given the word ‘transport’ is often used to describe motor stepping, we suggest ‘translocate’- also used in previous publications describing this assay (e.g. Kapitein et al., 2010; Kevenaar et al., 2016) – is a more appropriate term in the context of the peroxisome assay. We have changed this in the main text of the revised manuscript and rearranged the text to make it clearer what is being measured. In the analysis presented in Figure 5D,E, we performed a qualitative analysis in which we classified cells as either supporting translocation – when peroxisomes relocalise to the periphery – or not – when peroxisomes remained in the cell centre. From this, we calculated the percentage of cells in which translocation of peroxisomes was observed. We have now also added a more detailed description of our approach in the Materials and methods section.

B) Figure 5: I assume that the assay has been performed in interphase cells. KIF15 however does not localise to microtubules in interphase cells but rather to actin stress fibres (Buster et al., 2006), presumably by interaction with myosin 2B (Feng et al., 2016). Theoretically, some of the peroxisome behaviour observed here could therefore be actin/myosin dependent as well. This is probably not the case since the myosin 2B interaction was mapped to the KIF15-tail, which is missing here and the motor domain is probably overexpressed, uncoupling it form endogenous targeting/regulation mechanisms. However, to validate the assay in the first place, it should have been tested briefly, whether the KIF15 MD localises to microtubules at all.

Indeed, the peroxisome translocation assay was performed in interphase cells. Although the peroxisome assay was previously used to characterise a range of kinesins including KIF15 (Kevenaar et al., 2016), we have now specifically checked the localisation of our KIF15 construct. To do this, we expressed KIF15_MDC–HA–FRB and PEX3–mRFP–FKBP1 in COS cells, added rapalog to initiate kinesin translocation, fixed and immunostained with an anti-HA antibody to visualize the motor, and either an anti-tubulin antibody to visualise microtubules or phalloidin to visualize F-actin. These data show that KIF15_MDC predominantly localises along microtubules (Author response image 1, top panels), whereas almost no localisation on actin fibres or peripheral actin is observed (Author response image 1, bottom panels).

**Author response image 1. sa2fig1:** KIF15 localizes on microtubules in the peroxisome motility assay. (a, b) Representative images of COS-7 cells expressing KIF15_MDC–HA–FRB and PEX–mRFP–FKBP treated with rapalog for three hours and co-stained for (a) microtubules (rabbit anti-α-tubulin, 1:1000, Abcam) or (b) F-actin (Phalloidin Alexa647, 1:40, Life technologies). Zooms of the boxed regions are shown on the right. Scale bars, 10 µm.

C) Figure 5 D/F: As far as I understand it, the authors define an intensity threshold and quantify the cell area within 7.5 or 5 µm from the cell perimeter that is 'covered' by intensity larger a certain threshold. Why 5 µm in one case and 7.5 µm in the other case? Is thresholding really a good idea here? Given a fixed total-intensity-over-threshold equally distributed along the 5 µm (center to periphery) or with a distribution highly skewed towards the periphery. The skewed distribution would indicate better transport but would cover less area. Would something like fraction from overall intensity (above threshold) in a certain distance from the cell perimeter be more accurate?

We have applied the reviewer’s suggestion for analysis of our translocation data. While this approach does not alter our conclusions, we agree that the reviewer’s suggestion is more robust and we have used it to replace our previous analysis (Figure 5F,G (former Figure 5D,F)), and updated Figure 5 legend and the relevant Materials and methods section.

D) Please provide an explanation (e.g. in the legend or methods) for why the quantification thresholds are different for the two kinesins in Figure 5D/F. For Kif15 the periphery is defined as the outer 7.5μm of the cell, but for Kif1A it is 5μm. Does this threshold value affect the interpretation of the results?

We used different threshold values and defined different peripheral areas for the two kinesins KIF1A and KIF15 in quantification of the peroxisome assay because we observed that translocation of the peroxisomes by KIF15 into the periphery is less efficient within the three-hour timeframe of rapalog treatment. This is probably due to differences in KIF1A and KIF15 motor activity such that it takes longer for KIF15 to translocate a similar number of peroxisomes to the periphery compared to KIF1A. To capture the activity of KIF15 on peroxisome translocation, we adjusted the threshold intensity and defined a slightly larger peripheral area. In the revised manuscript, we have explained this thresholding and analysis approach in the relevant parts of the Results and Materials and methods. We have now also included images of peroxisome positioning in KIF1A_MDC-transfected cells (Figure 5—figure supplement 3) to allow direct comparison with the KIF15 examples already included in Figure 5C.

E) Figure 5: KBP inhibits KIF15 motor domains in a 1:1 stoichiometry. The authors kind of 'control' for the KBP expression (subsection “Cell biology image analysis and quantification”: 'Images were acquired of cells that express similar levels of HA-KBP constructs based on immunostaining'), but not for the KIF15/KIF1A-MD levels. KIF15/KIF1A-MD levels might change from mutant to mutant due to different stability in vivo (and from cell to cell, transient expression). Hence, we cannot formally exclude that a drop in peroxisome transport could as well be caused by [c] KBP > [c] KIF15 MD, masking positive (i.e., disruptive) hits or that false positive hits would be created when [c] KIF15 MD > [c] KBP.Thus, the authors should please provide a quantification of the KIF15-MD expression, allowing the reader to estimate, whether the KIF15/KBP ratios are the same throughout the experiment. Since the authors selected single cells based on the KBP signal, KIF15-MD quantification should also be done per cell.

In our experiments, we selected cells based on equivalent levels of KBP expression. When multiple constructs are transiently transfected, the relative expression levels of each construct does differ from cell to cell. Therefore, it is very likely that the selected cells express slightly different levels of motor constructs, as well as levels of PEX. Controlling for the precise expression levels of all three constructs on a cell to cell basis, would be very time-consuming, labour intensive and never perfect. We believe that the best way to control for such different expression levels is to analyse multiple cells from different experiments, which is why our analysis includes 28-35 cells per condition from two independent experiments. We agree with the reviewer that, in principle, the expression and activity of KIF15 and KIF1A motors may change from mutant to mutant due to different KBP stability. However, when we express different KBP mutants, we do not observe a major difference in the expression pattern or localization of the mutants compared to WT KBP, and have now provided additional data to demonstrate this (Figure 5—figure supplement 1). Therefore, we do not think that variable expression of KBP mutants will differentially affect motor expression, and conclude rather that the output of the assay reflects the KBP-motor interaction.

F) Figure 5—figure supplement 1: The authors compared the binding of WT and mutant KBP variants by quantifying the amount of HA-KBP that coprecipitated together with bioGFP-KIF15-MD. The amount is given as "intensity KBP-PD/intensity KBP input" normalised to the WT KBP condition. However, it is not clear whether the PD/Input quotient has been calculated first and then been normalised to WT KBP or if the quotient has been calculated from already normalized values. First case would not be ok, as Input and PD were clearly run and quantified on different gels and are therefore obviously not directly comparable – e.g., the pull-down bands for KBP to L10m in the first panels of (b) and (c) are stronger than their corresponding input bands.However, I did a quick quantification (ImageJ) of the blots shown in the left panel of (b) normalising each blot to WT KBP first. Still more protein was pulled down than has been put into the assay in the first place – how can this be? Also. Doing it like this, L12m would not be a positive hit any longer. While trying to quantify the other blots, I recognized that the tonal ranges for the KIF15-MD blots (b, right panel), (c, both panels) and the KBP input blot in (c, left) has massively been pushed to the higher intensities, leaving virtually no background in the lanes above and below the bands. If this happened during figure preparation – please don't do that. If this already happened due to wrong settings during data acquisition in the imager, your quantifications might not have bene accurate.I recommend to (re-)run both input and PD sample together on the same gel (there are 26-well gels commercially available). Like this they are treated the same way during western blotting/detection – allowing a more reliable cross comparison.G) Figure 5—figure supplement 1: comparing the PD/Input quotient only makes sense, if we assume that KIF15/ KIF1A-MD pulldown occurs from all lysates with the same efficiency and the overall KIF15/KIF1A-MD amount is not limiting KBP pulldown. This, however, is clearly not the case: in (c) lanes L14m, L16m, L18m, L12m+L14m and alphaHP4am the amount of pulled down KIF1A-MD is much less than for the other constructs, while the KBP input in comparison is even higher. Less KIF15/KIF1A-MD in the pulldown fraction might reflect a weaker KIF15/KIF1A-MD expression (mutant stability in vivo, transient transfection), less efficient biotinylation by BirA (transiently transfected as well) in vivo or less efficient pulldown by the streptavidin beads – we just can't tell from these figures. Nevertheless, if just 10% of the total KIF15/KIF1A-MD are pulled down, only 10% of total KBP should be expected to co-precipitate as well.Hence, the authors should please show and quantify the KIF15/KIF1A-MD input as well and relate the fraction of co-precipitated KBP with the fraction of pulled down motor domain (i.e., are mutants underperforming regarding their expected pulldown efficiency?).

Discussion points F and most of G from the reviewers are related. In order to address these concerns, we have re-run all the Western blots so that input and pull-down fractions are on the same gel to allow a more reliable cross comparison between conditions. In addition, we have changed the quantification method such that we now incorporate the amount of kinesin in the pull-down fraction in the calculation, since this affects the amount of KBP that can be pulled down. Thus, we now first quantify the intensity of KBP in the pull-down fraction divided by the intensity of KBP in the input fraction. This ratio is then divided by the intensity of the kinesin in the pull-down fraction and the obtained values are normalized to the KBP control. We emphasise that these intensity calculations were performed on raw data. On the images shown in the figure the contrast has been pushed up to generate clear images that are easy to interpret by the reader. We have updated the legend of Figure 5—figure supplement 2 to explain this.

In Author response image 2 we show all these western blot data and analysis. We propose to replace former Figure 5—figure supplement 1B (now Figure 5—figure supplement 2B) with KIF15 PD1, 2, 3 and 4 from N=2 and Figure 5—figure supplement 1C (now Figure 5—figure supplement 2C) with KIF1A PD5 and 6 from N=1, and PD7 and 8 from N=2 and to show the new quantification in this figure. However, if the reviewers/editors think all the new analysis should be included in the manuscript (i.e. the entirety of Author response image 2), we would be happy to follow their recommendation.

**Author response image 2. sa2fig2:** Overview of pull-down experimental data. Pull-down experiments showing the interaction between (a) KIF15_MDC or (b) KIF1A_MDC and mutated KBP constructs in HEK293T cell lysates. Two individual experiments for each pull-down are shown. Graphs on the right show the quantification of the intensity of the mutated HA-KBP construct in the pull-down fraction over the input fraction divided by the intensity of bioGFP_MDC in the pull-down fraction and normalized to HA-KBP. Quantifications of single experiments are shown as well as graphs combining the results from both experiments. Data are displayed as mean ± s.e.m. (data from two independent experiments).

I would also like to encourage the authors to verify their positive hits (i.e., L12m, L14m, HP4am, HP4bm and HP5m), by co-precipitating defined amounts of recombinant KBP/KIF-MD proteins to be on the safe side. Since expression was done in bacteria and purification protocols are established this could be done in a reasonable amount of time.

Due to COVID-19 restrictions in our labs, we are unfortunately not able to undertake these experiments in a reasonable amount of time. However, by following the other recommendations of the reviewers, the additional data and analyses we have incorporated render our conclusions robust.

H) In Figure 5—figure supplement 1C, the intensity of the bait (bioGFP-Kif1A_MDC) appears to be significantly lower for the L16m and L18m lanes compared to the other lanes. A lower level of bait could explain the weaker pulldown of HA-KBP in these lanes. The data should be re-quantified to take into account the amount of bait pulled down, or the experiment should be repeated with a consistent level of bait across all lanes (as in Figure 5—figure supplement 1). Since the authors highlight the weaker interaction between the L18m KBP and Kif1A compared to Kif15 in the text, this is an important control to include.

We agree with the reviewer that the amount of bait should be included in the quantification, and therefore we changed our quantification method (see response 11). Our western blot data show that L18m binds to both KIF1A_MDC as well as KIF15_MDC. This contradicts our results from the peroxisome assay, which indicates binding of L18m to KIF15, but not KIF1A. However, in our peroxisome assay with KIF1A we do observe that fewer cells have peroxisome translocation when L18m is expressed compared to control, suggesting that L18m may also has a weak binding to KIF1A. We have described these discrepancies more clearly as part of the text restructuring mentioned above (subsection “KBP binds kinesin motor domains via conserved motifs in the α-solenoid edge loops and α-helices at the concave face”).

I) Figure 5 and Figure 5—figure supplement 1: The L10m and L16m mutation bind KIF15-MD like WT KBP, but fail to inhibit peroxisome transport. If this effect is real (see points above), the authors should please comment on these results, since they are in conflict with their simple binding = inhibition model.J) The text doesn't address inconsistencies between the results of the peroxisome assay and the immunoprecipitation. For example, in Figure 5D/E, the mutation to loops 10 and 16 causes the peroxisomes to disperse to the periphery, which indicates that the inhibition of kinesin by KBP is relaxed. However, in Figure 5—figure supplement 1B the interaction between KBP and the kinesin does not appear to be significantly lower (and in the case of L10 is stronger than the control). These results appear to be directly contradictory. Can the authors provide an explanation for this effect?

We apologise for not properly addressing the examples of inconsistencies between the peroxisome assay and pull-down data in the previous version of our manuscript. The reanalysis of our data that we have undertaken highlights a general trend that the kinesin–KBP interaction is more susceptible to disruption in the context of the translocation assays compared to the pull-down assays. These presumably arise from the differences between cellular experiments and biochemical assays, which could lead to different outcomes when we look at binding properties of proteins. For example, this could be a function of the FRB–FKBP-mediated motor dimerization or might also reflect the fact that cellular MTs in the translocation assay can directly compete with KBP for kinesin binding. We have described these differences more clearly as part of the text restructuring mentioned above (subsection “KBP binds kinesin motor domains via conserved motifs in the α-solenoid edge loops and α-helices at the concave face”).